# A novel system for classifying tooth root phenotypes

**Jason Gellis**⬥*◉, **Robert Foley**◉

Department of Archaeology, The Leverhulme Centre for Human Evolutionary Studies, University of Cambridge, Cambridge, England

◉ These authors contributed equally to this work.
* jg760@cam.ac.uk

## Abstract

Human root and canal number and morphology are highly variable, and internal root canal form and count does not necessarily co-vary directly with external morphology. While several typologies and classifications have been developed to address individual components of teeth, there is a need for a comprehensive system, that captures internal and external root features across all teeth. Using CT scans, the external and internal root morphologies of a global sample of humans are analysed (n = 945). From this analysis a method of classification that captures external and internal root morphology in a way that is intuitive, reproducible, and defines the human phenotypic set is developed. Results provide a robust definition of modern human tooth root phenotypic diversity. The method is modular in nature, allowing for incorporation of past and future classification systems. Additionally, it provides a basis for analysing hominin root morphology in evolutionary, ecological, genetic, and developmental contexts.

## Introduction

Human dental morphology is a diverse collection of non-metric traits: cusp numbers, fissure and ridge patterns, root number and shape, and even congenital absence. Recording systems, such as the widely utilized Arizona State University Dental Anthropology System (ASUDAS) [1, 2], have been developed to catalogue these traits and their variants under a standardized scoring procedure; and to study how these variants are partitioned within and between populations. However, dental trait scoring systems are overwhelmingly focused on tooth crown morphology, with less attention paid to roots. Like tooth crowns, roots exhibit considerable variability in number, morphology, and size. For example, premolars have been reported as having between one to three roots [3, 4], while maxillary and mandibular molars have between one and five roots [5–8]. The literature has also long recognized several unusual morphological variants such as Tomes' root [9], taurodont roots [10], and C-shaped roots [11]. Additionally, the diversity of the canal system, both in number and configuration, has been an area of extensive study (Table 1).

As the number of catalogued external and internal morphologies grow, there is an increasing need for a comprehensive system, that can be used for documented and new morphotypes,

**Data Availability Statement:** CT scan data cannot be shared publicly because of ownership by Drs. Marta Miraźon-Lahr, Francis Rivera, and Lynn Copes. Some CT data are available from the University of Cambridge Duckworth Collection Institutional Data Access / Ethics Committee

(contact via http://www.human-evol.cam.ac.uk/duckworth.html) for researchers who meet the criteria for access to confidential data. The data underlying the results presented in the study are available from Jason Gellis - jg760@cam.ac.uk or The minimal anonymized data set necessary to replicate this study's findings are available at the Open Science Framework – • https://osf.io/9ynur/ • DOI 10.17605/OSF.IO/9YNUR.

**Funding:** The authors received no specific funding for this work.

**Competing interests:** The authors have declared that no competing interests exist.

and is robustly capable of describing the total human tooth root. The aim of this study is to 1) systematically describe the diverse internal and external morphologies of the human tooth root complex (i.e., all roots present in an individual tooth); and 2) define, develop, and provide a comprehensive system that captures these morphologies in all the teeth of both jaws for analysis.

## Background

The studies discussed below have addressed root number, canal number, external root morphology, canal morphology, and canal configuration independently. However, they comprise only parts of the tooth root complex, and thus provide a basis for a comprehensive phenotype system.

### Root number

Root number is probably the best studied element of root morphology, as counting roots is easily accomplished in extracted and in-situ teeth. Early studies of roots were primarily descriptive of root number, and the occasional metrical analysis [12–26]. Maxillary premolars are reported as having the most variation in number of roots, with a higher percentage of $P^3$s having two roots (or at least bifurcated apices), while $P^4$ is typified by one root. Three rooted maxillary premolars ($P^3$ and $P^4$) have been documented in modern humans [4, 17, 18, 20, 27, 28] but are extremely rare. Scott and Turner [2] report that Sub-Saharan Africans have the highest frequency at 65%, 40% in West Eurasian populations, 20–30% in East Asian populations, and 5–15% in Northeast Siberians and Native Americans. In contrast to the maxilla, the most frequent form of mandibular $P_3$s and $P_4$s is single rooted; though $P_3$s are occasionally two-rooted or, more rarely, thee rooted [29–31].

Maxillary molars are generally three rooted; though molars with two, four [32, 33] and five [34] roots have been reported. Variation in root number has been recorded for three rooted $M^2$s; with Australian Aboriginals having the highest reported percentage at 95.8% [15]. Sub-Saharan Africans also have a high frequency of three-rooted $M^2$s at 85%, Western Eurasians and East Asians ranging from 50–70% and American Arctic populations ranging from 35–40% [2]. Three European samples by Fabian, Hjelmmanm, and Visser (in [24]) report an average of 56.6%, in accordance with Scott and Turner [2]. Inuit populations are lower with East Greenland populations at 23.7% [22] and 30.7–31.3% in two prehistoric Alaskan populations [35].

**Table 1. Previous typological studies of tooth roots and canals in modern humans.**

| Authors | Technique | Roots | Canals | Teeth |
|---|---|---|---|---|
| Tomes [9] | Direct observation | Yes | - | Premolars |
| Keith [10] | Direct observation | Yes | Yes | Molars |
| Ackerman et al., [70] | Radiography | Yes | Yes | Molars |
| Vertucci et al., [55] | Direct observation using dye | - | Yes | Maxillary premolars |
| Abbot [71] | Direct observation, radiography | Yes | Yes | All teeth, focus on premolars |
| Turner et al., [1] | Direct observation | Yes | - | All |
| Carlsen and Alexandersen [72] | Direct observation | Yes | - | Mandibular molars |
| Hsu and Kim [57] | Sectioning of tooth, direct observation using dye | - | Yes | Maxillary and mandibular pre- and first molars. |
| Fan et al., [11] | Radiography | Yes | Yes | 2nd mandibular molar |
| Moore et al., [60] | CT | Yes | Yes | Premolars |
| Ahmed et al., [56] | micro CT | - | Yes | All |

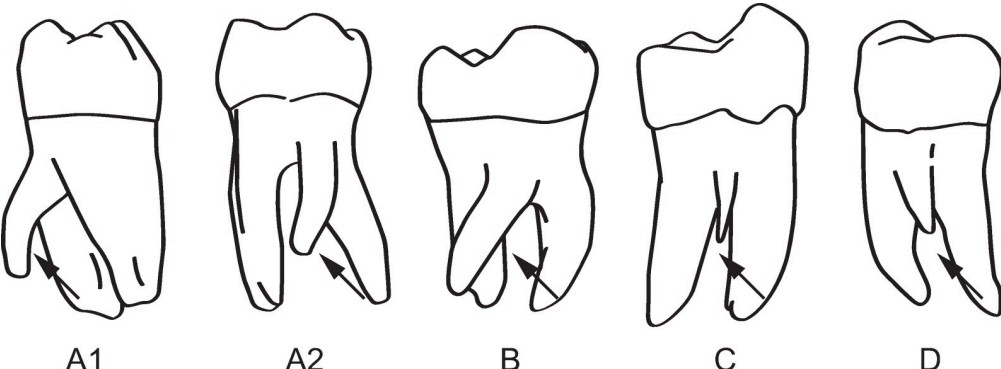

**Fig 1. Examples of accessory roots.** Mandibular 1st molars with A1 and A2: radix entomolaris (left = distolingual surface, right = lingual surface), B: M3 with radix entomolaris (lingual view), C: M1 with furcation root (buccal view), D: M1 with fused radix paramolaris (buccal view). Modified from Calbersen et al. [36].

Unlike their maxillary counterparts, mandibular molars are less variable in root number. A rare exception, mandibular molars sometimes exhibit a third accessory root (Fig 1). They are generally smaller than the mesial and distal roots of the mandibular molars, and most frequently appear in lower first molar. In the ASUDAS these are referred to as three rooted molars [35]. The clinical literature applies a different typology and identifies several variants. These include–(1) The radix entomolaris (En) accessory root arising from the lingual surface of the distal root; (2) the radix paramolaris (Pa) arising from buccal side of the distal root; and (3), furcation root (Fu) projecting from the point of bifurcation between roots [36].

The entomolaris trait is expressed with high frequency (20–25%) in Sino-American populations (East Asia, North East Siberia, American artic), with one Aleut population exhibiting a sample frequency of 50% [2, 37]. The trait also appears in 15.6% of North American Athabascans and Algonquin Native American tribes [38]. Tratman [19] claimed the trait showed a distinct dichotomy between European and Asian populations, as did Pedersen [22]. Comparatively, this trait appears in less than 1% of populations from Sub-Saharan Africa, West Eurasia, and New Guinea (ibid). The trait has been reported in extinct hominins [39], but see Scott et al. [40], for a further discussion.

Single rooted molars usually appear in three forms: C-shaped $M_2$s, taurodont $M_1$s-$M_3$s, and pegged $M^3$s/$M_3$s (Fig 2A–2C). C-shaped molars are common in Chinese populations with a frequency as high as 40% [41]. The trait has a low frequency of 0–10% in Sub-Saharan Africans [42], 1.7% in Australian Aboriginals [15], and 4.4% in the Bantu (Shaw, 1931). Rare in modern humans, taurodont molars occur when the root trunk and internal pulp cavity are enlarged and apically displaced. This form was first classified by Keith [10] in *Homo neanderthalensis*. Externally, taurodont roots are cylindrical in shape (Fig 2B). While sometimes confused with C-shaped molars, taurodont roots lack an internal and external 180° arc, and are instead circular in cross-section, usually with a bifid apical third. Pegged third molars are the most variable in size and morphology [35]. Their reduction has a genetic component and patterned geographical variation [35, 43]. Pegged third molar roots are associated with a reduced crown, appear more frequently in the maxilla than the mandible, and are circular in cross-section.

Multi-rooted anterior teeth are exceptionally rare [44]. Alexandersen [45] compiled data on double rooted mandibular canines from several European countries, two Danish Neolithic samples, and two medieval samples; in which they attain a frequency of 4.9–10%. His findings suggest that the double rooted canine trait is a European marker. However, Lee and Scott [46] found the variant in 1–4% of an East Asian population sample (Central Plains China, Western

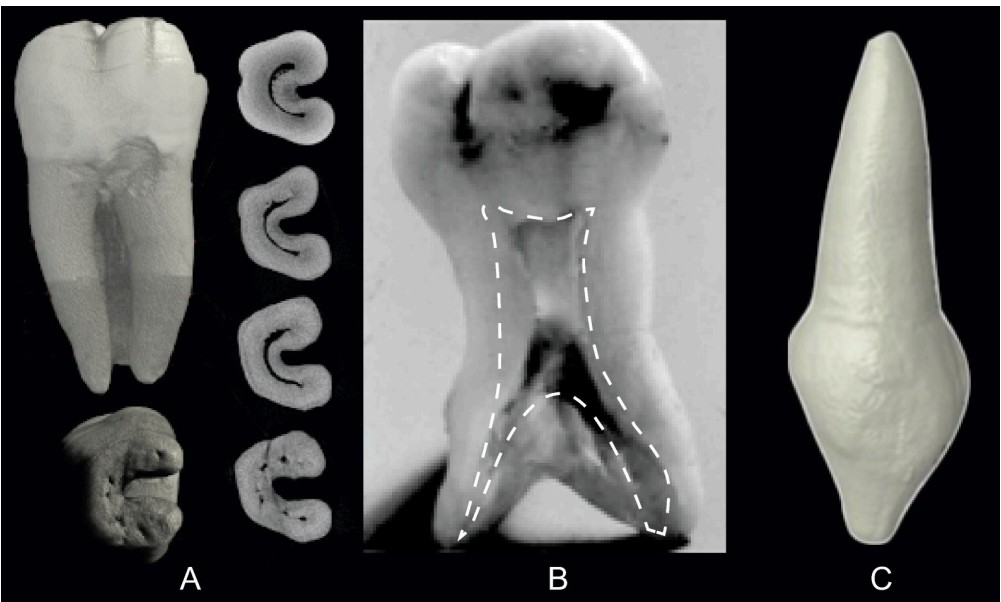

**Fig 2. Unusual root forms.** A. C-shaped tooth in (clockwise from top left) lingual, cross-section at the cemento-enamel junction, cervical third, middle third, apical third, and apical views. B. Taurodont molar, apically displaced pulp chamber and canals outlined in white. C. Peg-shaped root. Images A, C from the Root Canal Anatomy Project http://rootcanalanatomy.blogspot.com/ (accessed 10 March 2021). Image B from http://www.dentagama.com (accessed 27th March 2021).

China and Mongolia, Northern China, Ordos Region, and Southern China). The authors interpreted this as possible evidence of an eastward migration of Indo-European speaking groups into China and Mongolia.

### External root morphology

Studies of root morphologies in cross-section have recognised forms such as 'plate-like' and 'dumb-bell', in the mandibular molars of humans, great apes, cercopithecoids, and Plio-Pleistocene hominins [47–50]; while cross sections of australopith anterior teeth have been described as 'ovoid' [51]. While these descriptions appear from time to time in the literature (ibid), they are inconsistently applied, have not been described in detail required for comparative studies, or codified into a classification system which can be consistently applied.

Some exceptions exist. Tomes' roots [9] have a long history of study in the anthropological literature, and are included in the ASUDAS. These single rooted teeth are part of a morphological continuum in which the mesial surface of the root displays, in varying degrees of depth, a prominent developmental groove [1]. Tomes first described this root configuration in modern human mandibular premolars and classified it as a deviation from the "normal" European single rooted premolar (ibid). Tomes' root appears in 10% of $P_3$s and $P_4$s of the Pecos Native American Tribe [18], 36.9% of $P_3$s and 8.4% for $P_4$s in the Bantu [17], and >25% for Sub-Saharan African groups [2]. In contrast, $P_3$ Tomes' roots account for 0–10% of Western Eurasian populations and 10–15% of North and East Asian population (ibid). In its most extreme form, the groove appears on mesial and distal surface, and can result in bifurcation of the root. In cross section, Tomes' roots have a V-shaped 'notch' where the two radicals are dividing. Occasionally, this division results in bifid apices or two separate roots, depending on the level of bifurcation [50].

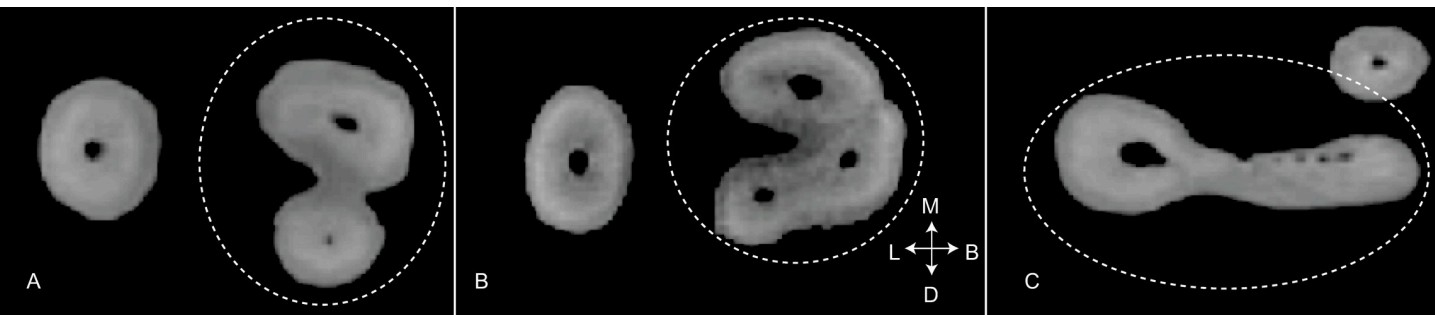

**Fig 3. Fusion of multiple roots into right single roots in maxillary 2nd molars.** A. fused mesial elliptical (E) and distal globular (G) root types. B. fused mesial E and distal hourglass (H) root types. C. fused lingual G and distal plate (P) root types. Images A, B, and C from the Root Canal Anatomy Project https://rootcanalanatomy. blogspot.com/ (accessed 10 March 2020).

Another unusual morphology, the C-shaped molar (Fig 2A) consists of a single root in an 180° arc, with a buccally oriented convex edge, and are most common in the 2nd mandibular molar [8, 41]. In certain cases, two mandibular molar roots are fused on their buccal side giving them the appearance of C-shaped molars; however, the two forms are not homologous and can be discerned by the former's lack of a uniform, convex external buccal surface, and C-shaped canal.

Occasionally two roots can become fused (Fig 3). The reasons for fusion are unclear, but may be due to suppression or incomplete fusion of the developing tooth root's interradicular processes during root formation [52, 53]. Fused roots can be joined by dentine, have linked pulp chambers and/or canals [54]. In such a scenario adjacent root structures are apparent, but their separation is incomplete. Fused roots are most common in the post-canine tooth row of the maxillary arch.

## Root canals

In its simplest form, a root canal resembles a tapered cylinder, extending from the pulp chamber beneath the crown, and exiting the root apex. Often, individual canals are circular or ovoid in cross section, even when multiple canals appear in the same root (Fig 4).

With exception of the anterior teeth, this is rarely the case, and there are often multiple canal configurations within a single root (Fig 5). A wide range of canal configurations have been reported [11, 55–57], though the number of configurations vary by study. Often, these discrepancies are due to the inclusion or exclusion of accessory canals (lateral and furcation canals), which branch from the main canal structure, like the roots of a tree, at the point of bifurcation or the apical third. However, most practitioners opt to exclude these from typologies as they are not continuous structures from the pulp chamber to the root apex.

In conjunction with external form, canal morphology has proven useful for hominin classification [58–61]. In mandibular premolars, researchers have shown that combined external morphologies and canal configurations can differentiate robust and gracile australopiths [50, 60, 62]. However, it is unclear how internal variation relates to external morphology or is partitioned between and across populations.

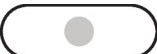 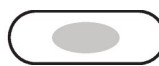 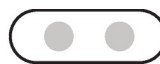 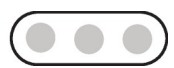

**Fig 4. Canal morphologies in cross section.** Left to right: Round and ovoid canal forms. Gray is canal shape, black is external form of the tooth root.

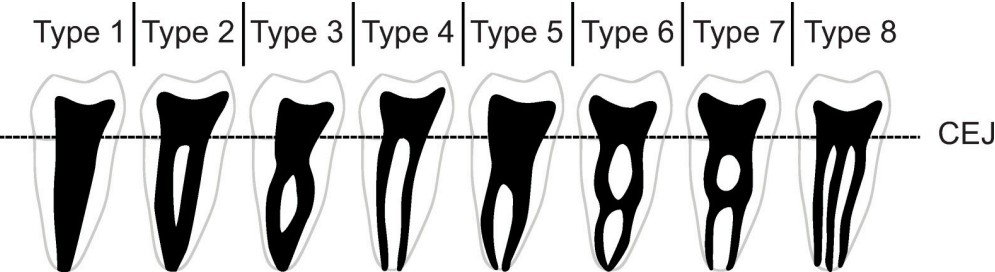

**Fig 5. Vertucci's widely used canal classification system.** Root and canal number do not always conform to one another. Black area represents pulp chambers and various canal configurations [55].

## Canal classification systems

The most widely used canal typology system contains eight types (Fig 5), which can, theoretically, be found in any tooth in the jaws [55]. However, this classification system does not include all known canal types. For example, canal isthmuses—complete or incomplete connections between two round canals are frequently found in molars (Fig 6, left), though they appear in other roots [57]. These canal configurations are distinct from those described by Vertucci et al. [55]. Likewise, C-shaped canals have been the subject of several studies [8, 63, 64], and their configurations are nearly identical (though ordered differently) to the canal isthmuses described by Hsu and Kim [57], only stretched around an 180° arc (Fig 6, right). These same isthmus canal configurations can also be found in Tomes' roots [65, 66].

Many classification systems have been introduced (Table 1). However, they often focus on one tooth type or morphology. Of the 27 traits catalogued by the ASUDAS, only root number for specific teeth ($P^3$, $M^2$, $C_1$, $M_1$, and $M_2$) and external morphology (Tomes' root) are included [1]. Others systems are only focused on the canal configurations of maxillary premolars [33], the mesial canals of mandibular 1st and 2nd molars [67], or more narrowly, unusual canal types such as isthmus [57] or C-shaped canals [11, 64]. Others propose separate classificatory nomenclature based on root number [56], or maxillary [68] and mandibular molars [69].

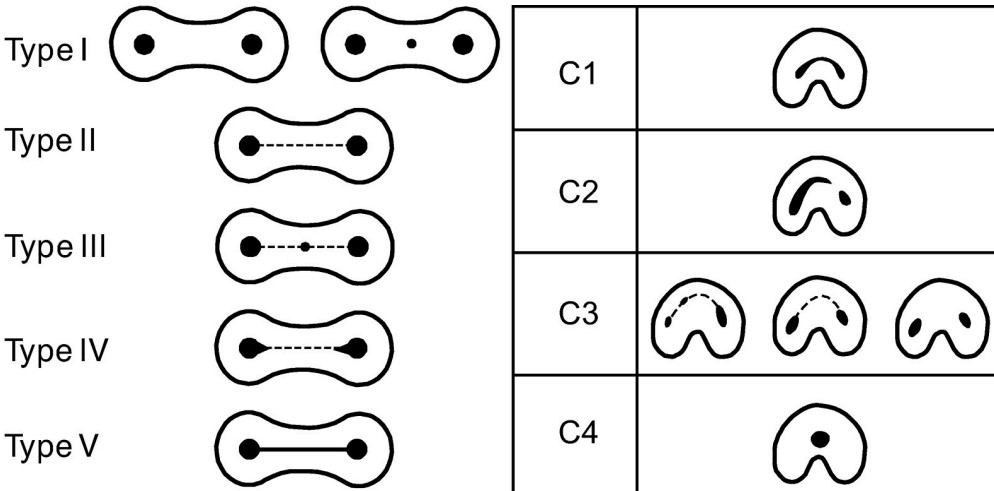

**Fig 6. Two different classification methods for canal isthmuses.** Left: Canal isthmuses, modified from Hsu and Kim [57]. Right: C-shaped root canals, modified from Fan et al., [11].

While canal number and morphology do not always conform to external number and morphology [50, 54, 60], the literature on the relationship between internal and external morphologies is sometimes inconsistent. For example, Vertucci *et al.* [55] categorize maxillary premolars with two separate canals as type IV (Fig 6). However, it is unclear if this classification is to be applied only to two canals encased in a single or two-rooted tooth. Canal shape can also change over time due to dentin deposition [73]. While some variation may be due to age and/or biomechanical factors, there is currently no methodology to classify these changes.

## Materials and methods

### CT scans

Using cone-beam computed tomography (CBCT or CT), both sides of the maxillary and mandibular dental arcades of individuals (n = 945) were analysed from osteological collections housed at the Smithsonian National Museum of Natural History (SI), American Museum of Natural History (AMNH), and the Duckworth Collection (DC) at the University of Cambridge Leverhulme Centre for Human Evolutionary Studies. Full skulls of individuals from the SI and AMNH were scanned by Dr. Lynn Copes [74] using a Siemens Somatom spiral scanner (70 μA, 110 kV, slice thickness 1.0 mm, reconstruction 0.5 mm, voxel size mm^3: 1.0x1.0x0.3676). Full skulls of individuals from the DC were scanned by Professor Marta Miraźon-Lahr and Dr. Frances Rivera [75] using a Siemens Somatom Definition Flash scanner at Addenbrooke's Hospital, Cambridge England (80μA, 120kV, slice thickness 0.6mm, voxel size mm^3: 0.3906x0.3906x0.3). There is a 0.1 mm difference between the two samples in terms of slice thickness, but studies have suggested that this is not a large enough difference to influence the results [54]. In collecting the data no discernible difference in data quality was evident for the authors of this study. For all collections, crania and mandibles were oriented on the rotation stage, with the coronal plane orthogonal to the x-ray source and detector. Permission to use the scans has been granted by Dr. Copes, Professor Miraźon-Lahr and Dr. Rivera. A complete list and description of individuals included in this study is listed in the supporting information (S1 Table).

Transverse CT cross sections of roots and canals were assessed in the coronal, axial, and sagittal planes across the CT stack, using measurement tools in the Horos Project Dicom Viewer (Fig 7) version 3.5.5 [76]. Only permanent teeth with completely developed roots and closed root apices were used for this study. While information for all teeth from both sides of the maxillary and mandibular arcades was recorded, only the right sides were used to avoid issues with asymmetry and artificially inflated sample size.

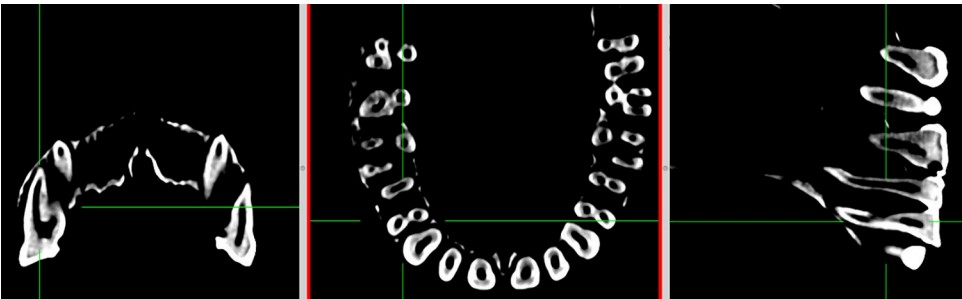

**Fig 7. Horos Dicom Viewer 2D orthogonal view used to assess root and canal morphologies.** Left: Coronal view at mid-point of roots. Centre: Anterior view at midpoint of roots. Right: Lateral view at midpoint of roots.

## Anatomical descriptions

Categorically, incisors are indicated by an I, canines a C, premolars with P, and molars use M. Tooth numbers are labelled with super- and subscripts to differentiate the teeth of the maxilla and mandible. For example, $M^1$ indicates the 1st maxillary molar while $M_1$ indicates the 1st mandibular molar. Numerically, incisors are numbered either 1 or 2 for central and lateral incisors respectively. Canines are marked 1 as there exists only one canine in each quadrant of the jaws. Through the course of evolution, apes and old world monkeys have lost the first and second premolars of their evolutionary ancestors, thus the remaining 2 premolars are numbered 3 and 4 [77, 78].

Unlike the anatomical surfaces and directions used for tooth crowns, there exists no formula for tooth roots. However, classical anatomical terms–mesial, buccal, distal, lingual, or combinations of (e.g., mesio-buccal), can be used to describe the location of roots and canals. Additionally, the term axial is used to describe a single or centrally located canal within a single-rooted tooth. Because anatomical location rather than anatomical surface is being employed, buccal replaces labial (for anterior teeth) when describing roots.

## Exclusion criteria and observer error

A number of teeth and/or individuals were excluded from this study due to: damaged/broken teeth, poor CT scan quality, and CT artifacts (e.g., beam hardening, rings, cupping, etc). Skulls and/or mandibles that were misaligned/poorly aligned to the anatomical plane might distort morphologies (e.g., a round canal may appear oval shaped if the CT slice is at an oblique angle). This problem is mitigated by visualizing the morphology in question across multiple planes through the CT stack, and as a CT volume. However, for cases in which the researcher felt unsure, these individuals were excluded from analysis. An observer error test was conducted for the tooth root morphologies described and analysed below (S2 Table). Results show a high degree of accuracy between a trained (study author) and untrained observer for identifying and recording tooth root internal and external morphologies.

## External root morphology

External root morphology was assessed at the measured mid-point of the root, bounded by the cemento-enamel junction (CEJ) and root apex/apices. The midpoint was chosen as a point of inspection because (a) the root has extended far enough from the CEJ, and in the case of multi-rooted teeth, from the neighboring roots to be structurally and developmentally distinct [79]; and (b) at a point in the eruptive phase in which the adjoined tooth crown is in functional occlusion [80]; and, (c) does not reflect the morphological alteration common to the penetrative phase in which the apical third of the root becomes roughened and/or suffers ankylosis or concrescence due to penetration of the bones of the jaws [80].

## Root and canal number

To determine root and canal number, the Turner Index [1], which compares the point of root bifurcation (POB) relative to total root length, was applied. When this ratio is greater than 33% of the total root or canal length, the root or canal is classified as multi-rooted. When the ratio is less than 33% the root or canal is considered single rooted, or with a bifid apical third (Fig 8). Here, a single root canal is defined as a canal which extends from the pulp chamber within the crown and exits at a single foramen. Accessory canals are not included in this study.

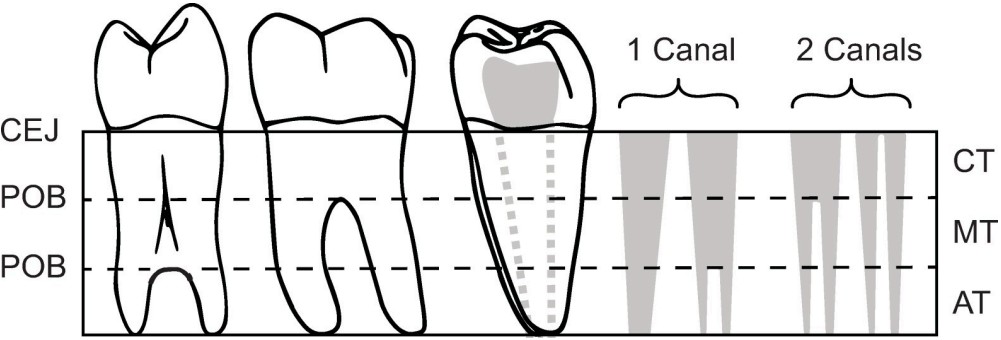

**Fig 8. Determination of root and canal number.** Left = Distal view of single-rooted premolar with bifurcation of the apical third of the root. Middle: Lingual view of double-rooted mandibular molar. Right: Distal root of double-rooted mandibular molar with examples of canal counts in solid gray. Dotted gray lines indicate canal/s position in root. CEJ = Cemento-enamel junction, POB = Point of bifurcation, Solid gray = canals. CT = cervical third, MT = middle third, AT = apical third.

## Canal morphology and configuration

Individual canals are circular or ovoid in cross section. Here, circular or round canals are classified as R, and ovoid canals as O. These are appended numerically to reflect the number of canals present. For example, R2 simply describes two, distinct circular canals, while O describes a single ovoid canal (Fig 9).

For classification, canal configurations have been simplified into five categories, R-R5, that reflect canal number and account for fusion/division of canals (Fig 10). These categories can be found in any tooth and are applied to single roots within the root complex (e.g., 3 roots, each with a single canal, would not be designated R3, but R for each canal per root).

Because C-shaped canal configurations [11] are nearly identical to the canal isthmuses described by Hsu and Kim [57] (Fig 6), both isthmus and C-shaped canal systems were combined and simplified into one (Fig 11). Five categories are described for canal isthmuses. Here, i1 is defined as a single root with two unconnected canals (here classified as R2, Figs 9 and 10); i2 is defined as a complete connection between separate canals; i3 is defined by one or both canals extending into the isthmus area, but without complete connection; i4 is defined by an incomplete connection between three (sometimes incomplete) canals; and i5 is defined as a thin or sparse connection between two canals. These same isthmus canal configurations can also be found in Tomes' roots.

## Results

CT scans of 5,970 teeth (Table 2) of 945 individuals from a global sample (S1 and S2 Tables) were analysed to identify morphologies which are useful for classifying the tooth root complex of modern human teeth. Descriptive statistics of external and internal morphologies are presented (Tables 2–5 and 7–10). From the external and internal morphologies, a novel tooth root classification system comprised of phenotype elements, each of which describes a property of the individual roots, and the root complex as a whole, is defined and explained. Each element

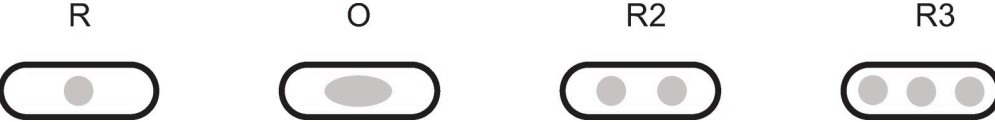

**Fig 9. Canal morphologies in cross section.** Gray is canal shape, black is external form of the tooth root.

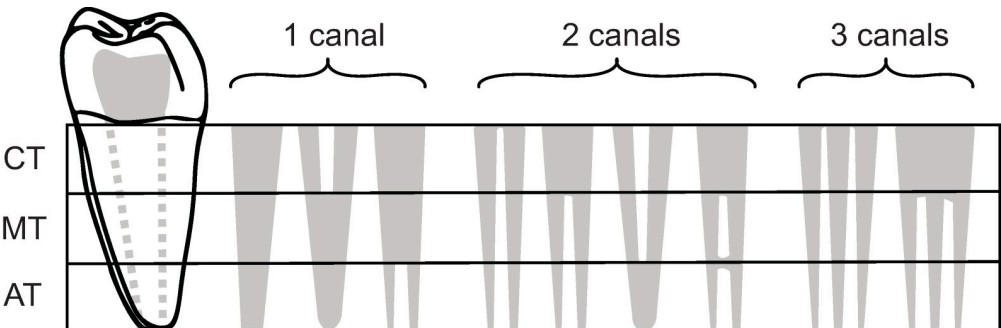

**Fig 10. Canal counts and degrees of separation.** Solid grey = root canal forms. CT = cervical third, MT = middle third, AT = apical third.

(E) within the set provides information on root (E1) and canal (E2) number; identification and location of roots and canals in the root complex (E3); external root form (E4); and (E5) internal canal forms and configurations. Combined elements (for example root number and internal canal form combined together) can be treated as phenotypes or separated and analysed by their constituent parts. The system, described below, allows us to define a finite set of possible root phenotypes and their permutations (the realized phenotypic set) and analyse diversity in a constrained morphospace.

## Root number

In aggregate, the number of roots in teeth from the sample are between one and four (Table 3). Anterior teeth almost always having a single root, the exception being two mandibular canines, premolars between one and three roots, and molars between one and four roots. Entomolaris, or three-rooted molars, appear in 18.05% $M_1$s, 1.23% of $M_2$s, and 5.94% of $M_3$s, while paramolaris appears in 3.63% of $M_3$s.

## Canal number

Teeth in this study contain between one and six canals (Table 4), and it is not uncommon for a single root to contain two or more canals, especially in the molars. With the exception of $I^1$, all

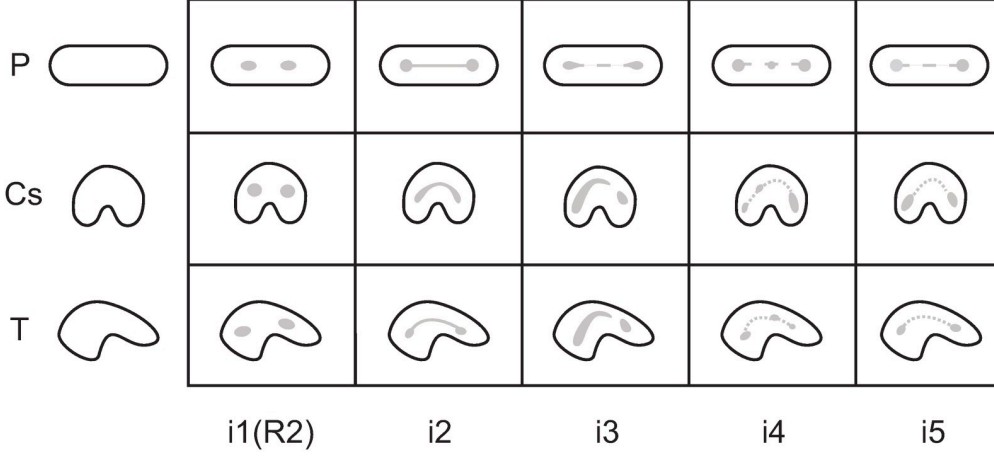

**Fig 11. Combined isthmus classification system.** Based on systems developed by Hsu and Kim [57] and Fan et al. [11]. Black is external root form and grey is canal form. P = plate-shaped, Cs = C-shaped, and T = Tomes' root.

**Table 2. Tooth counts of the right side of the maxillary and mandibular dental arcades.**

| Tooth | n | Tooth | n | Total |
|---|---|---|---|---|
| Maxilla | | Mandible | | |
| $I^1$ | 204 | $I_1$ | 204 | 408 |
| $I^2$ | 248 | $I_2$ | 247 | 495 |
| $I^2$ CON | 1 | $I^2$ CON | 1 | - |
| $C^1$ | 406 | $C_1$ | 295 | 701 |
| $P^3$ | 515 | $P_3$ | 343 | 858 |
| $P^4$ | 467 | $P_4$ | 313 | 780 |
| $M^1$ | 697 | $M_1$ | 410 | 1,107 |
| $M^2$ | 596 | $M_2$ | 385 | 981 |
| $M^3$ | 362 | $M_3$ | 278 | 640 |
| $M^3$ CON | 28 | $M^3$ CON | 25 | - |
| Total | 3,495 | - | 2,475 | 5,970 |

Superscript = maxilla, subscript = mandible. I = incisor, C = canine, P = premolar, M = molar. CON = congenitally absent teeth (discussed in section 1.5).

**Table 3. Number of roots in teeth of the maxilla and mandible by tooth.**

| Tooth | Root number | n | Total Roots | % of teeth* | Tooth | Root number | n | Total Roots | % of teeth* |
|---|---|---|---|---|---|---|---|---|---|
| Maxilla | | | | | Mandible | | | | |
| $I^1$ | 1 | 204 | 204 | 100.00 | $I_1$ | 1 | 204 | 204 | 100.00 |
| $I^2$ | 1 | 248 | 248 | 100.00 | $I_2$ | 1 | 247 | 247 | 100.00 |
| $C^1$ | 1 | 406 | 406 | 100.00 | $C_1$ | 1 | 293 | 297 | 99.32 |
| | | | | | | 2 | 2 | | 0.68 |
| $P^3$ | 1 | 295 | 739 | 57.28 | $P_3$ | 1 | 341 | 345 | 99.42 |
| | 2 | 216 | | 41.94 | | 2 | 2 | | 0.58 |
| | 3 | 4 | | 0.78 | | | | | |
| $P^4$ | 1 | 405 | 530 | 86.72 | $P_4$ | 1 | 313 | 313 | 100.0 |
| | 2 | 61 | | 13.06 | | | | | |
| | 3 | 1 | | 0.22 | | | | | |
| $M^1$ | 1 | 2 | 2,060 | 0.29 | $M_1$ | 2 | 336 | 894 | 81.95 |
| | 2 | 28 | | 4.02 | | 3En | 74 | | 18.05 |
| | 3 | 666 | | 95.55 | | | | | |
| | 4 | 1 | | 0.14 | | | | | |
| $M^2$ | 1 | 56 | 1,561 | 9.39 | $M_2$ | 1 | 49 | 727 | 12.73 |
| | 2 | 117 | | 19.63 | | 2 | 330 | | 85.71 |
| | 3 | 421 | | 70.64 | | 3 | 1 | | 0.26 |
| | 4 | 2 | | 0.34 | | 3En | 5 | | 1.30 |
| $M^3$ | 1 | 89 | 831 | 24.59 | $M_3$ | 1 | 20 | 563 | 7.19 |
| | 2 | 82 | | 22.65 | | 2 | 231 | | 83.09 |
| | 3 | 186 | | 51.38 | | 3En | 16 | | 5.76 |
| | 4 | 5 | | 1.38 | | 3Pa | 11 | | 3.96 |

* From Table 2. Congenitally absent teeth not included in statistics for this table.

En = Entomolaris, Pa = Paramolaris.

**Table 4. Number of canals per tooth in the maxilla and mandible by tooth.**

| Tooth | Canal number | n | Total Canals | % of teeth* | Tooth | Canal number | n | Total Canals | % of teeth* |
|---|---|---|---|---|---|---|---|---|---|
| | | Maxilla | | | | | Mandible | | |
| $I^1$ | 1 | 204 | 204 | 100.00 | $I_1$ | 1 | 180 | 228 | 88.24 |
| | | | | | | 2 | 24 | | 11.76 |
| $I^2$ | 1 | 247 | 249 | 99.60 | $I_2$ | 1 | 208 | 286 | 84.21 |
| | 2 | 1 | | 0.40 | | 2 | 39 | | 15.79 |
| $C^1$ | 1 | 405 | 407 | 99.75 | $C_1$ | 1 | 273 | 317 | 92.54 |
| | 2 | 1 | | 0.25 | | 2 | 22 | | 7.46 |
| $P^3$ | 1 | 82 | 959 | 15.92 | $P_3$ | 1 | 254 | 435 | 74.05 |
| | 2 | 422 | | 81.94 | | 2 | 86 | | 25.07 |
| | 3 | 11 | | 2.14 | | 3 | 3 | | 0.87 |
| $P^4$ | 1 | 233 | 708 | 49.89 | $P_4$ | 1 | 300 | 326 | 95.85 |
| | 2 | 228 | | 48.82 | | | | | |
| | 3 | 5 | | 1.07 | | 2 | 13 | | 4.15 |
| | 4 | 1 | | 0.22 | | | | | |
| $M^1$ | 2 | 4 | 2,431 | 0.57 | $M_1$ | 2 | 27 | 1,431 | 6.59 |
| | 3 | 355 | | 50.93 | | | | | |
| | 4 | 333 | | 47.78 | | 3 | 167 | | 40.73 |
| | 5 | 4 | | 0.57 | | 4 | 205 | | 50.00 |
| | 6 | 1 | | 0.14 | | 5 | 10 | | 2.44 |
| | | | | | | 6 | 1 | | 0.24 |
| $M^2$ | 1 | 8 | 1,910 | 1.34 | $M_2$ | 1 | 2 | 1,107 | 0.52 |
| | 2 | 21 | | 3.52 | | 2 | 93 | | 24.16 |
| | 3 | 408 | | 68.46 | | 3 | 241 | | 62.60 |
| | 4 | 159 | | 26.68 | | 4 | 49 | | 12.73 |
| $M^3$ | 1 | 32 | 1,065 | 8.84 | $M_3$ | 1 | 10 | 748 | 3.60 |
| | 2 | 24 | | 6.63 | | 2 | 86 | | 30.94 |
| | 3 | 239 | | 66.02 | | 3 | 162 | | 58.27 |
| | 4 | 67 | | 18.51 | | 4 | 20 | | 7.19 |

* From Table 2. Congenitally absent teeth not included in statistics for this table.

single rooted anterior teeth have a double canaled variant. Molars have the most canals per tooth, with $M^1$s showing the most variation. With the exception of $I^1$, canal number exceeds root number (Table 3).

## Anatomical orientation of canals in the root complex

The majority of teeth follow a similar anatomical pattern of having axially (A) oriented, buccal (B) and lingually (L) oriented, or mesially (M), distally (D), and lingually (L) oriented canals and roots. Other orientations, for example MB1DB1ML1DL1R, are relatively rare, and only appear in molars. In cases where there are multiple canals appear in a single root these are almost always found in the mesial or buccal orientations (e.g., M2D1L1, B2L1).

## External root morphology at midpoint

Similar to the variation found in tooth cusp morphology [1], external root morphologies exist as distinctive, yet easily recognizable anatomical variants (Fig 12). While these morphologies frequently extend through the apical third to the apex of the root, occasionally they are bifid (Bi), and have been noted this where applicable.

**Table 5. Anatomical orientation of the canals in the maxilla and mandible by tooth.**

| Tooth | External morphology | n | % of teeth* | Tooth | External morphology | n | % of teeth* |
|---|---|---|---|---|---|---|---|
| | Maxilla | | | | Mandible | | |
| I¹ | A | 204 | 100.00 | I₁ | A | 180 | 88.24 |
| | | | | | B1L1 | 24 | 11.76 |
| I² | A | 247 | 99.60 | I₂ | A | 208 | 84.21 |
| | B1L1 | 1 | 0.40 | | B1L1 | 39 | 15.79 |
| C¹ | A | 405 | 99.75 | C₁ | A | 273 | 92.54 |
| | B1L1 | 1 | 0.25 | | B1L1 | 22 | 7.46 |
| P³ | A | 82 | 15.92 | P₃ | A | 254 | 74.05 |
| | | | | | B1L1 | 86 | 25.07 |
| | B1L1 | 421 | 81.75 | | M1D1L1 | 3 | 0.87 |
| | B1L2 | 1 | 0.19 | | | | |
| | B2L1 | 6 | 1.17 | | | | |
| | M1D1 | 1 | 0.19 | | | | |
| | M1D1L1 | 4 | 0.78 | | | | |
| P⁴ | A | 233 | 49.89 | P₄ | A | 300 | 95.85 |
| | B1L1 | 228 | 48.82 | | B1L1 | 13 | 4.15 |
| | B2L1 | 3 | 0.65 | | | | |
| | B2L2 | 1 | 0.21 | | | | |
| | M1D1L1 | 2 | 0.43 | | | | |
| M¹ | B1L1 | 3 | 0.43 | M₁ | M1D1 | 27 | 6.59 |
| | M1D1 | 1 | 0.14 | | M1D1L1 | 12 | 2.93 |
| | M1D1L1 | 354 | 50.80 | | M2D1 | 156 | 38.05 |
| | M1D1L2 | 2 | 0.29 | | M2D1L1 | 59 | 14.39 |
| | M1D2 | 1 | 0.14 | | M2D2 | 144 | 35.12 |
| | M1D2L1 | 1 | 0.14 | | M2D2L1 | 3 | 0.73 |
| | M1L1 | 1 | 0.14 | | M2D3 | 5 | 1.22 |
| | M2D1 | 1 | 0.14 | | M3D1 | 1 | 0.24 |
| | M2D1L1 | 327 | 47.07 | | M3D2 | 2 | 0.49 |
| | M2D1L2 | 2 | 0.29 | | M3D3 | 1 | 0.24 |
| | M2D2L1 | 1 | 0.14 | | | | |
| | M2D2L2 | 1 | 0.14 | | | | |
| | M3D1L1 | 1 | 0.14 | | | | |
| | MB1DB1ML1DL1 | 1 | 0.14 | | | | |
| M² | A | 8 | 1.34 | M₂ | A | 2 | |
| | B1L1 | 20 | 3.36 | | B1D1L1 | 1 | |
| | M1B1D1L1 | 1 | 0.17 | | B2L1 | 2 | |
| | | | | | B2L2 | 1 | |
| | M1D1 | 1 | 0.17 | | M1B1D1 | 7 | |
| | M1D1L1 | 408 | 68.46 | | M1D1 | 93 | |
| | M1D2L1 | 2 | 0.33 | | M1D1L1 | 1 | |
| | M2D1L1 | 153 | 25.67 | | M1D2 | 1 | |
| | MB1DB1ML1DL11 | 2 | 0.33 | | M2D1 | 229 | |
| | ML3D1 | 1 | 0.17 | | M2D1L1 | 4 | |
| | | | | | M2D2 | 42 | |
| | | | | | M3D1 | 2 | |

(*Continued*)

**Table 5.** (Continued)

| Tooth | External morphology | n | % of teeth* | Tooth | External morphology | n | % of teeth* |
|---|---|---|---|---|---|---|---|
| M$^3$ | A | 32 | 8.84 | M$_3$ | A | 10 | |
| | B1L1 | 17 | 4.70 | | B2L1 | 1 | |
| | B2D1L1 | 1 | 0.28 | | M1B1D1 | 8 | |
| | M1B1D1L1 | 2 | 0.55 | | M1B2D1 | 1 | |
| | M1D1 | 7 | 1.93 | | M1D1 | 84 | |
| | M1D1L1 | 235 | 64.92 | | M1D1L1 | 9 | |
| | M1D1L2 | 1 | 0.28 | | M2B1D1 | 2 | |
| | M1D2 | 3 | 0.83 | | M2D1 | 147 | |
| | M1D2L1 | 1 | 0.28 | | M2D1L1 | 9 | |
| | | | | | M2D2 | 6 | |
| | M2D1 | 1 | 0.28 | | M3D1 | 1 | |
| | M2D1L1 | 45 | 12.43 | | | | |
| | M2D2 | 11 | 3.04 | | | | |
| | MB1DB1ML1DL1R | 5 | 1.38 | | | | |
| | ML3D1 | 1 | 0.28 | | | | |

* From Table 2. Congenitally absent teeth not included in statistics for this table. A = axial, M = mesial, B = Buccal, D = Distal, L = Lingual.

**Table 6. Description of external tooth root morphologies at the midpoint.**

| Morphology | Description | Reference |
|---|---|---|
| Globular (G) | Round or circular in shape. While this form varies greatly in size, it is relatively invariant in shape, and in that all edges are relatively equidistant from the center. | [81] |
| Elliptical (E) | While size, and distance of the edges from the center vary, elliptical shaped roots are distinct from others in that they look like a squashed circle. Sometimes these forms are perfectly symmetrical and other times they resemble and egg. However, a consistent feature are there continuously smooth edges which are concentric to the canals. | [60, 80, 81] |
| Wedge (W) | Wedge shaped roots are easily distinguished by their 'tapered' appearance. Sometimes these forms take the shape of a triangle with three edges and corners, while other times they appear more teardrop shaped with a slight constriction in the middle. However, they are easily distinguished as the buccal end is always noticeably wider than the lingual end. | This study |
| Hourglass (H) | Hourglass shaped roots have often been confused with plate-shaped roots, or occasionally, elliptical roots. However, this form is distinct and easily identified by its bulbous ends and constricted center. This constriction can be so pronounced that the root appears almost as a lemniscate in cross-section. | This study, but see [47–49] for complimentary and contradictory definitions. |
| Kidney (K) | Kidney shaped roots are defined by their opposite convex and concave sides. Sometimes these curvatures are pronounced, and other times they are more subtle. However, these two features are always apparent, and distinct from other forms. | This study |
| Plate (P) | Plate shape roots are similar to hourglass and elliptical roots in their dimensions but are easily distinguished by their flat edges. In some variants the corners are rounded, while in others they are square. | This study, but see [47–49] for complimentary and contradictory definitions. |
| Tomes' (T) | Tomes' roots have been documented for nearly a century and appear in a number of classification systems including the ASUDAS. They are single rooted teeth that appear to be 'splitting' into two roots. In cross section they sometimes look like c-shaped molar roots. However, one of their distinguishing features is that they are only found in mandibular premolars. | [1, 9] |
| C-shaped (CS) | C-shape molars are primarily found in the second molars of the mandible, though they rarely appear in the first and third mandibular molars as well. There is a substantial clinical literature covering their distinct morphology and prevalence. Unlike Tomes' roots they do not appear to be splitting into two roots. Rather, they are a single, continuous root structure. Like kidney shaped roots they have opposite convex and concave sides. However, their curvature is more pronounced, in nearly a 180° arc with ends that are parallel to one another. | [8, 11, 82] |

**Table 7. Number of external root morphologies in the maxilla and mandible by tooth.**

| Tooth | External morphology | n | % of roots* | Tooth | External morphology | n | % of roots* |
|---|---|---|---|---|---|---|---|
| | | Maxilla | | | | Mandible | |
| I[1] | E | 69 | 33.82 | I[1] | E | 13 | 6.37 |
| | G | 117 | 57.35 | | G | 1 | 0.49 |
| | | | | | K | 3 | 1.47 |
| | P | 8 | 3.92 | | P | 177 | 86.76 |
| | W | 10 | 4.91 | | W | 10 | 4.90 |
| I[2] | E | 120 | 48.39 | I[2] | E | 7 | 2.83 |
| | | | | | H | 1 | 0.40 |
| | G | 25 | 10.08 | | K | 10 | 4.05 |
| | P | 97 | 39.11 | | P | 219 | 88.66 |
| | W | 6 | 2.42 | | W | 10 | 4.05 |
| C[1] | E | 149 | 36.70 | C[1] | E | 54 | 18.18 |
| | Ebi[†] | 1 | 0.25 | | G | 6 | 2.02 |
| | G | 4 | 0.99 | | H | 6 | 2.02 |
| | P | 135 | 33.25 | | K | 2 | 0.67 |
| | | | | | P | 141 | 47.47 |
| | | | | | W | 87 | 29.29 |
| | W | 117 | 28.83 | | Wbi | 1 | 0.34 |
| P[3] | E | 10 | 1.35 | P[3] | E | 62 | 17.97 |
| | G | 402 | 54.40 | | G | 14 | 4.06 |
| | H | 80 | 10.83 | | H | 1 | 0.29 |
| | Hbi | 40 | 5.41 | | K | 3 | 0.87 |
| | K | 38 | 5.14 | | P | 145 | 42.03 |
| | | | | | T | 75 | 21.74 |
| | Kbi | 5 | 0.68 | | Tbi | 8 | 2.32 |
| | P | 143 | 19.35 | | W | 37 | 10.72 |
| | Pbi | 12 | 1.62 | | | | |
| | W | 9 | 1.22 | | | | |
| P[4] | E | 24 | 4.53 | P[4] | E | 122 | 38.98 |
| | G | 106 | 20.00 | | G | 21 | 6.71 |
| | H | 70 | 13.21 | | Hbi | 1 | 0.32 |
| | Hbi | 21 | 3.96 | | K | 1 | 0.32 |
| | K | 31 | 5.85 | | P | 155 | 49.52 |
| | Kbi | 3 | 0.57 | | T | 9 | 2.88 |
| | P | 266 | 50.19 | | Tbi | 1 | 0.32 |
| | Pbi | 4 | 0.75 | | W | 3 | 0.96 |
| | W | 4 | 0.75 | | | | |
| M[1] | E | 500 | 24.27 | M[1] | E | 20 | 2.24 |
| | G | 266 | 12.91 | | G | 76 | 8.50 |
| | H | 11 | 0.53 | | H | 188 | 21.03 |
| | K | 49 | 2.38 | | Hbi | 61 | 6.82 |
| | | | | | K | 73 | 8.17 |
| | P | 668 | 32.43 | | Kbi | 4 | 0.45 |
| | Pbi | 4 | 0.19 | | P | 437 | 48.88 |
| | W | 536 | 26.02 | | Pbi | 17 | 1.90 |
| | Wbi | 2 | 0.09 | | W | 18 | 2.01 |

(*Continued*)

**Table 7.** (Continued)

| Tooth | External morphology | n | % of roots* | Tooth | External morphology | n | % of roots* |
|---|---|---|---|---|---|---|---|
| M² | E | 451 | 28.89 | M₂ | CS | 33 | 4.54 |
| | G | 371 | 23.76 | | CSBi | 1 | 0.14 |
| | H | 9 | 0.58 | | E | 33 | 4.54 |
| | | | | | G | 15 | 2.06 |
| | Hbi | 2 | 0.13 | | H | 143 | 19.67 |
| | K | 80 | 5.12 | | Hbi | 17 | 2.34 |
| | Kbi | 1 | 0.06 | | K | 206 | 28.34 |
| | | | | | Kbi | 4 | 0.55 |
| | P | 262 | 16.78 | | P | 256 | 35.21 |
| | W | 241 | 15.43 | | Pbi | 5 | 0.69 |
| | | | | | W | 1 | 0.14 |
| M³ | E | 105 | 12.64 | M₃ | CS | 6 | 1.07 |
| | G | 338 | 40.67 | | E | 75 | 13.32 |
| | | | | | G | 72 | 12.79 |
| | H | 12 | 1.44 | | H | 49 | 8.70 |
| | Hbi | 5 | 0.60 | | Hbi | 4 | 0.71 |
| | | | | | K | 182 | 32.33 |
| | K | 41 | 4.93 | | Kbi | 3 | 0.53 |
| | | | | | P | 155 | 27.53 |
| | P | 115 | 13.84 | | Pbi | 2 | 0.36 |
| | Pbi | 5 | 0.60 | | W | 4 | 0.71 |
| | W | 103 | 12.39 | | | | |

* from Table 3, **Bi** = bifid. Congenitally absent teeth not included in statistics for this table.

Though some of these morphologies have been discussed in the literature, their descriptions are inconsistent (e.g., hourglass, plate). Table 6 includes definitions and descriptions of the root morphologies shown in Fig 12. Two of the morphologies, wedge (W) and kidney (K), are described here for the first time.

External root morphologies appear in different frequencies in each tooth, and some morphologies do not appear in some teeth at all (Table 7). The number of morphologies increase posteriorly along the tooth row, and $M_1$s have the most morphologies. Part of this is due to the number of bifid (Bi) variants (e.g., EBi, PBi, etc.), as well as the presence of pegged and fused roots (Tables 8 and 9, respectively).

Pegged (Mi) roots while globular in cross section, are therefore considered their own distinct morphology as they are a form of microdontia [83]. They are relatively rare in this sample and only appear in $M^3$ and $M_3$ (Table 8).

Fused roots are almost always found in the molars and are more common in the maxillary molars (Table 9). In almost all cases fusion includes the mesial (M) root, and it is not uncommon for fused roots to have some degree of bifurcation (Bi).

**Table 8. Type and number of teeth with pegged roots.**

| Tooth | External morphology | n | % of roots* | Tooth | External morphology | n | % of roots* |
|---|---|---|---|---|---|---|---|
| | Maxilla | | | | Mandible | | |
| M³ | Mi | 5 | 0.60 | M₃ | Mi | 6 | 1.07 |

* from Table 3

**Table 9. Type and number of roots showing fusion morphologies.**

| Tooth | External morphology | n | % of roots* | Tooth | External morphology | n | % of roots* |
|---|---|---|---|---|---|---|---|
| | Maxilla | | | | Mandible | | |
| $P^4$ | MLFBi | 1 | 0.19 | $M_2$ | MDF | 13 | 1.79 |
| $M^1$ | MDF | 4 | 0.19 | $M_3$ | MDF | 5 | 0.89 |
| | MLF | 1 | 0.05 | | | | |
| | MLFBi | 3 | 0.15 | | | | |
| | DLF | 8 | 0.39 | | | | |
| | DLFBi | 8 | 0.39 | | | | |
| $M^2$ | BLF | 3 | 0.19 | | | | |
| | DLF | 8 | 0.51 | | | | |
| | DLFBi | 2 | 0.13 | | | | |
| | MDF | 12 | 0.77 | | | | |
| | MDFDLF | 4 | 0.26 | | | | |
| | MDFMLF | 2 | 0.13 | | | | |
| | MDFMLFBi | 1 | 0.06 | | | | |
| | MLF | 60 | 3.84 | | | | |
| | MLFBi | 21 | 1.35 | | | | |
| | MLFBiDLF | 1 | 0.06 | | | | |
| | MLFBiMDF | 1 | 0.06 | | | | |
| | MLFDLF | 23 | 1.47 | | | | |
| | MLFDLFBi | 3 | 0.19 | | | | |
| | MLFMDF | 4 | 0.26 | | | | |
| $M^3$ | DLF | 15 | 1.81 | | | | |
| | MDF | 14 | 1.68 | | | | |
| | MDFDLF | 1 | 0.12 | | | | |
| | MLF | 25 | 3.01 | | | | |
| | MLFBi | 8 | 0.96 | | | | |
| | MLFBiDLF | 2 | 0.24 | | | | |
| | MLFDLF | 36 | 4.33 | | | | |
| | MLFMDF | 1 | 0.12 | | | | |

* from Table 3. M = Mesial, B = Buccal, D = Distal, L = Lingual, F = Fused, Bi = bifid apex. Ex: MLF = mesio-lingual fused roots, MLFBi = mesio-lingual fused roots with bifurcation. Congenitally absent teeth not included in statistics for this table.

## Canal shape and configuration

Single round (R) and ovoid (O) canals are the most common canal morphologies and configurations in nearly all teeth of both jaws (Table 10). Interestingly, R canals are most prevalent in maxillary teeth while O canals are most prevalent in mandibular teeth. Isthmus canals (i2-i5) appear with less frequency than single (R and O) and double-canaled (R2-R5) variants and are mostly found in the mandibular molars. The double-canaled R5 orientation appears the least. No R3 variants appear in this sample.

## Classification system

As discussed in the literature review above, the categorization of roots and canals can be misleading or inaccurate when systems devised for a particular tooth type (e.g., premolar) are applied across other tooth types. Here, a new system that is simple, accurate, human and computer readable, and allows for easy qualitative and/or quantitative analysis of the entire

**Table 10. Number of canal shapes and configurations in the maxilla and mandible by tooth.**

| Tooth | Canal morphology | n~ | % of canals† | Tooth | Canal morphology | n~ | # of canals† |
|---|---|---|---|---|---|---|---|
| | | | Maxilla | | | | Mandible |
| $I^1$ | O | 22 | 10.78 | $I_1$ | O | 110 | 48.25 |
| | R | 182 | 89.22 | | R | 70 | 30.70 |
| | | | | | R2 | 5 | 4.39 |
| | | | | | R4 | 18 | 15.79 |
| | | | | | i2 | 1 | 0.88 |
| $I^2$ | O | 84 | 33.73 | $I_2$ | O | 140 | 56.68 |
| | R | 163 | 65.46 | | R | 68 | 27.54 |
| | R4 | 1 | 0.80 | | R2 | 6 | 2.43 |
| | | | | | R4 | 31 | 12.55 |
| | | | | | i2 | 1 | 0.40 |
| | | | | | i5 | 1 | 0.40 |
| $C^1$ | O | 227 | 55.77 | $C_1$ | O | 204 | 64.35 |
| | R | 178 | 43.74 | | R | 73 | 23.03 |
| | R5 | 1 | 0.49 | | R2 | 1 | 0.63 |
| | | | | | R4 | 15 | 9.46 |
| | | | | | R5 | 1 | 0.63 |
| | | | | | i2 | 3 | 1.89 |
| $P^3$ | O | 67 | 6.99 | $P_3$ | O | 177 | 40.69 |
| | R | 458 | 47.76 | | R | 84 | 19.31 |
| | | | | | R2 | 21 | 9.66 |
| | R2 | 120 | 25.03 | | R4 | 1 | 0.46 |
| | R4 | 75 | 15.64 | | i2 | 2 | 0.92 |
| | | | | | i3 | 1 | 0.46 |
| | R5 | 7 | 1.46 | | i4 | 2 | 1.38 |
| | i2 | 6 | 1.25 | | i5 | 59 | 27.13 |
| | i5 | 9 | 1.88 | | | | |
| $P^4$ | O | 193 | 27.26 | $P_4$ | O | 179 | 54.91 |
| | R | 163 | 23.02 | | R | 121 | 37.12 |
| | R2 | 70 | 19.77 | | R2 | 5 | 3.07 |
| | R4 | 90 | 25.42 | | R4 | 1 | 0.61 |
| | R5 | 3 | 0.85 | | i5 | 7 | 4.29 |
| | i2 | 11 | 3.11 | | | | |
| | i5 | 2 | 0.56 | | | | |
| $M^1$ | O | 357 | 14.69 | $M_1$ | O | 225 | 15.83 |
| | | | | | R | 142 | 9.99 |
| | R | 1,379 | 56.75 | | R2 | 261 | 36.73 |
| | | | | | R4 | 86 | 12.10 |
| | R2 | 149 | 12.26 | | R5 | 5 | 0.70 |
| | R4 | 134 | 11.03 | | i2 | 105 | 14.78 |
| | R5 | 14 | 1.15 | | i3 | 30 | 4.22 |
| | | | | | i4 | 10 | 1.41 |
| | i2 | 33 | 2.72 | | i5 | 30 | 4.22 |
| | i3 | 3 | 0.25 | | | | |
| | i4 | 1 | 0.08 | | | | |
| | i5 | 13 | 1.07 | | | | |

(*Continued*)

**Table 10.** (Continued)

| Tooth | Canal morphology | n~ | % of canals† | Tooth | Canal morphology | n~ | # of canals† |
|---|---|---|---|---|---|---|---|
| M² | O | 284 | 14.87 | M₂ | O | 295 | 26.66 |
| | R | 1,245 | 65.18 | | R | 90 | 8.13 |
| | R2 | 53 | 5.55 | | R2 | 139 | 25.11 |
| | R4 | 69 | 7.23 | | R4 | 99 | 17.89 |
| | | | | | R5 | 2 | 0.36 |
| | R5 | 4 | 0.42 | | i2 | 68 | 12.29 |
| | i2 | 45 | 4.71 | | i3 | 22 | 3.97 |
| | i3 | 7 | 0.73 | | i4 | 12 | 3.25 |
| | i4 | 1 | 0.16 | | i5 | 13 | 2.35 |
| | i5 | 11 | 1.15 | | | | |
| M³ | O | 120 | 11.27 | M₃ | O | 202 | 27.01 |
| | R | 740 | 69.48 | | R | 185 | 24.73 |
| | R2 | 44 | 8.26 | | R2 | 58 | 15.51 |
| | R4 | 25 | 4.69 | | R4 | 72 | 19.25 |
| | i2 | 21 | 3.94 | | i2 | 31 | 8.29 |
| | i3 | 3 | 0.56 | | i3 | 5 | 1.34 |
| | i4 | 1 | 0.28 | | i4 | 1 | 0.40 |
| | i5 | 8 | 1.50 | | i5 | 13 | 3.48 |

~ n column list times each variant appears. However, R2, R4, R5, and i2-i5 are two-canaled variants and are counted twice to calculate % of canals.

† = Table 4. Congenitally absent teeth not included in statistics for this table.

phenotype, or each of its constituent parts, individually or in any combination, is presented. Five phenotypic elements (E) that comprise the human tooth root phenotype have been outlined: E1—root number, E2—canal number, E3 –canal location, E4—external morphology, and E5 –canal morphology and configuration. The system provides codes for each element, and the resulting combination constitutes that root complex's complete phenotype code.

**Tooth name or number.** This system works with categorical and numbering systems including, but not limited to, the Palmer Notation Numbering system, the FDI World Dental Federation System, simple abbreviations such as UP4 (upper 2$^{nd}$ premolar) or LM1 (lower first molar), or the super- and subscript formulas described in and used throughout this study.

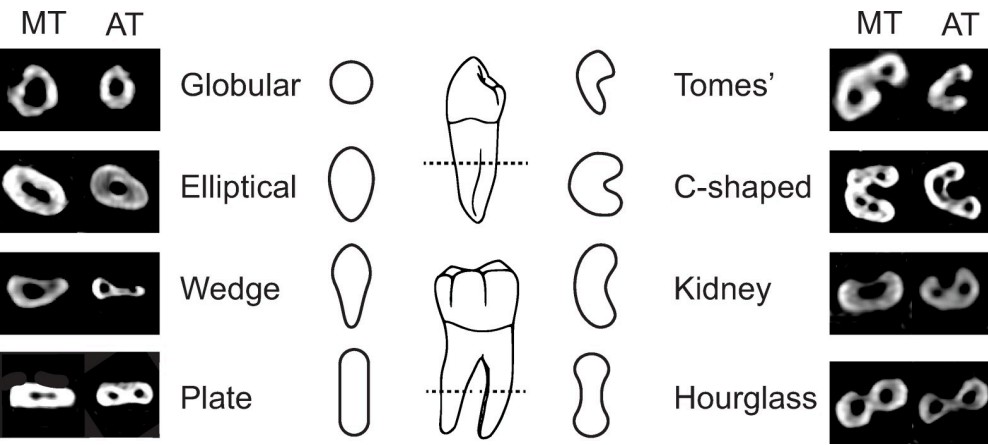

**Fig 12. External root morphologies.** Left and right columns = axial CT slices showing external root morphologies at the middle third (MT) and apical third (AT). Centre illustrations = root morphologies at centre of root/s.

**Root number or absence.** Roots are recorded by simple counts and represented with an R. For example, a two-rooted tooth would be coded as R2. Root number is determined using the Turner index [1] as outlined in the methods section. Congenitally absent teeth and roots are labeled CON, rather than 0 or NA. This is because congenital absence of a tooth is a heritable phenotypic trait, with different population frequencies [84, 85]. In the case of missing teeth, root number can often be recorded by counting the alveolar sockets. Fig 13 presents a workflow for recording E1 and its variants.

**Canal number.** Like root number, canal number is a simple count but represented with a C rather than an R. As discussed in the methods section, the Turner index (1991), essentially a system of thirds, was applied to determine counts. Building the above example, a two rooted, three canaled tooth would be coded as R2-C3. Fig 14 presents a workflow for recording E2 and its variants.

**Anatomical locations of canals.** The locations of the canals in the root complex are easily recorded following the anatomical directions common to any dental anatomy textbook and discussed above. Fig 15 presents a workflow for recording E3 and its variants. Labeling order begins with mesial (M), followed by buccal (B), distal (D), and lingual (L), inclusive of

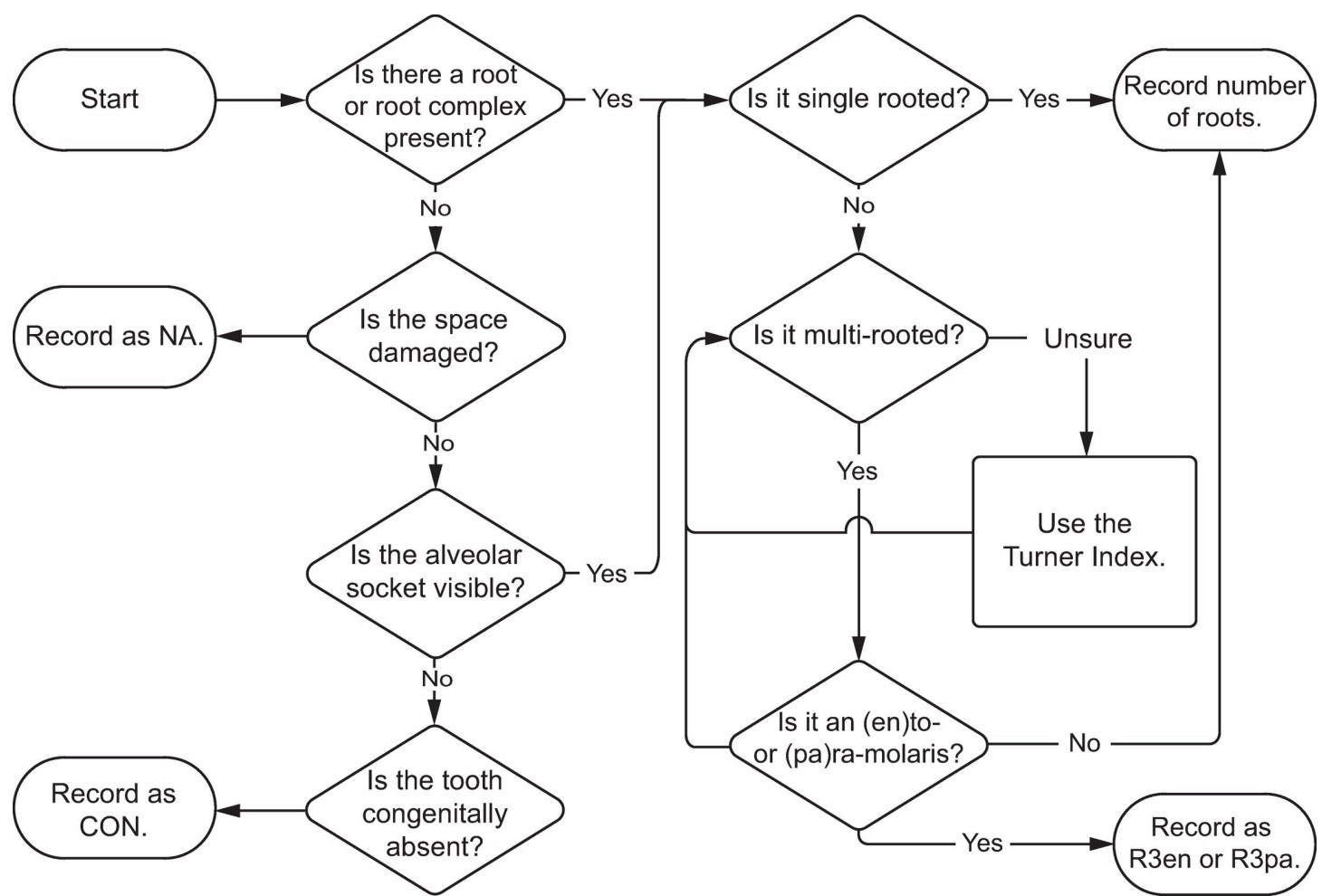

**Fig 13. Flow chart for determining and recording phenotype element 1—root number or absence.**

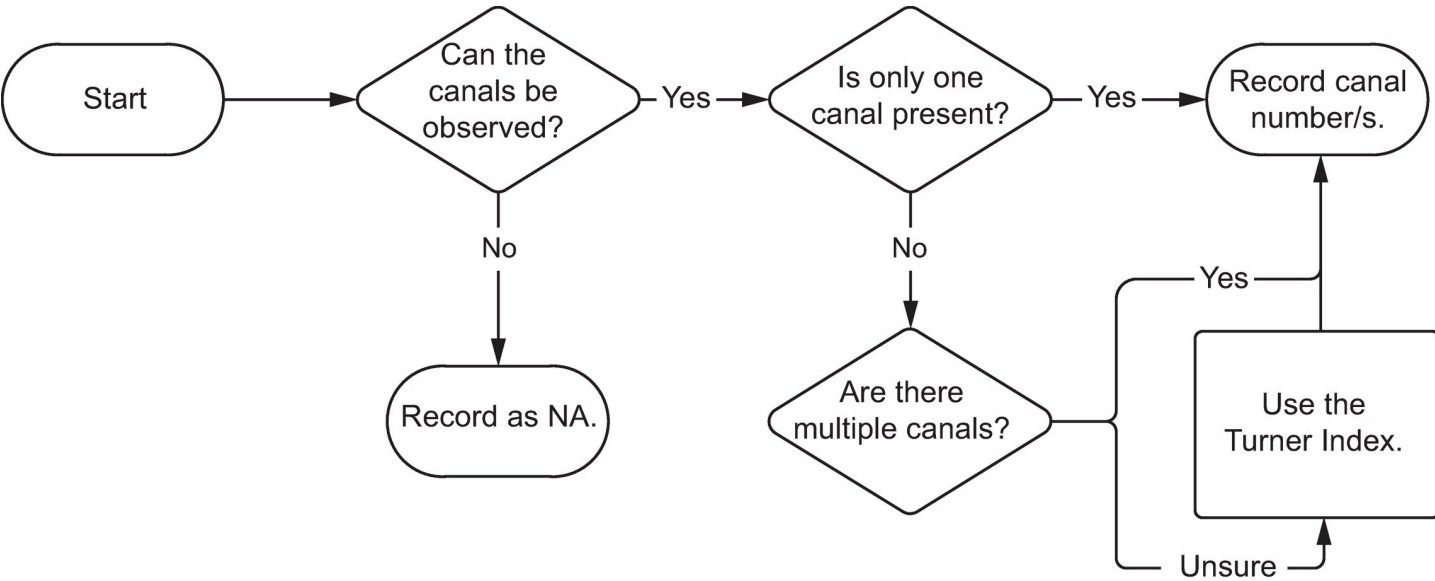

**Fig 14. Flow chart for determining and recording phenotype element 2—canal number.**

intermediate locations (e.g., mesio-distal). Continuing the above example, if two canals are found in the mesial root and one in the distal root, the root complex would be coded as R2-C3-M2D1.

**External root morphology.** Fig 12 and Table 6 visualize and describe external root morphologies recorded at the midpoint of the root, while Fig 16 presents a workflow for recording E4 and its variants. Fused roots also fall under E4. However, unlike the morphologies

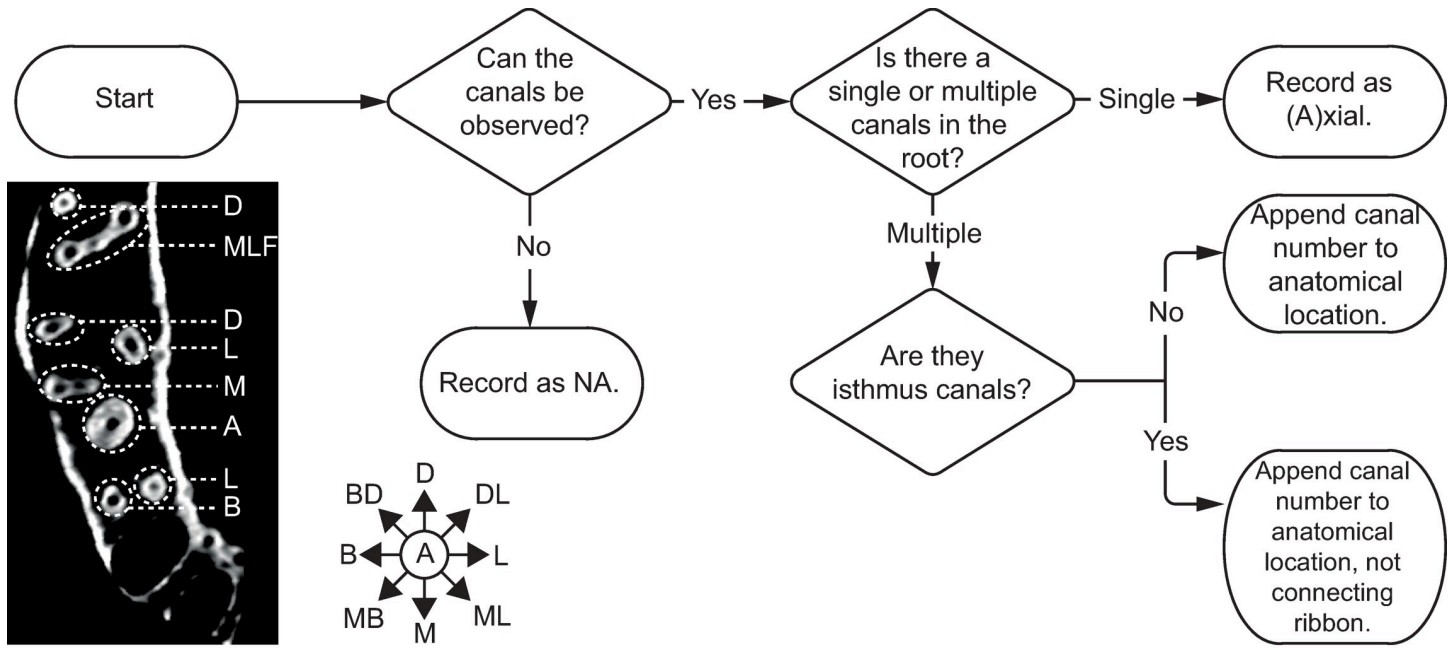

**Fig 15. Flow chart for determining and recording phenotype element 3—anatomical location of canals.** Bottom left: Axial CT scan slice of right maxillary dental arcade. Anatomical directions: A = axial, M = mesial, MB = mesio-buccal, B = buccal, BD = bucco-distal, D = distal, DL = disto-lingual, L = lingual, ML = mesio-lingual, F = fused.

described in Table 6 and Fig 12, fused roots are simply recorded with F (for fused) appended to the anatomical locations of the fused roots. For example, a mesial and buccal fused root, would be recorded as MBF. Though axial slices were used to determine these morphologies, morphologies can also be ascertained visually from extracted teeth, and occasionally the alveolar sockets of missing teeth [2]. A tooth with two roots, containing three canals–two in the mesial root and one in the distal root, with an hourglass and plate shaped mesial and distal roots, is coded as: R2-C3-M2D1-MHDP.

**Canal configuration.** Root canal configuration requires visualization of the canal system from the CEJ to the foramen/foramina. While µCT or CBCT provide the greatest resolution for visualising these structures, in certain cases 2D radiography is sufficient (see Versiani et al., 2018 for an indepth discussion and comparison of techniques). This simplified system (Figs 10 and 11) will help the user to classify accurately canal configurations as it is based on a system of thirds, rather than harder to visualize 'types'.

Figs 17 and 18 present a workflow for recording E5 and its variants. Finalizing the above example—two round canals in the mesial root and one ovoid canal in the distal root can easily be coded as MR2DO; completing the root complex phenotype code as: R2-C3-M2D1-MHDP-MR2DO (Fig 19).

## The phenotypic set within the morphospace of root diversity

Within these phenotypic elements, there is exist 874 unique phenotype element permutations derived from the global sample. These comprise this study's "phenotypic set" among the range of potential phenotypic permutations. Anterior teeth have the least number of permutations while molars, particularly maxillary molars, have the greatest (Fig 20).

## Discussion

This paper set out to present a method that would capture quantitatively and qualitatively the diversity of human tooth root phenotypes, using a modular approach. It has shown that it is possible to have a universal code for phenotyping roots, and that a global sample of modern humans demonstrates the high level of tooth root phenotype diversity. A more comprehensive set of tooth root data should reinforce and expand the broader toolset for studying human phenotypic diversity (e.g., tooth crowns, craniofacial morphometrics, genetics, etc.).

The large number of phenotypes permutations found in the sample can be explained by the variation within each element. For example, Table 11 shows how permutations in one phenotypic element can result in four nearly identical tooth roots with four different phenotype codes. Here, all these roots are identical in their phenotypic elements with the exception of their external morphology (E4). Teeth with more roots result in a greater number of permutations. Fig 21 illustrates how increasing numbers and multiple combinations, and orientations of root morphologies create the morphological permutations of the external phenotypic elements. However, compared to tooth crowns, the number of phenotype permutations is relatively few, as a recent test of ASUDAS crown traits indicates greater than 1.4 million combinations, or permutations of crown phenotypes [86].

Two elements of the approach are emphasized. The first is the expansion of data available and the use of a universal and modular system. Scanning technologies have provided greater access to tissues, such as tooth roots, that were previously difficult to access for visual inspection, thus, permitting a much fuller and complete description of these morphologies. The system was developed and designed to be comprehensive and universal, so that any tooth can be placed within the set of attributes. The five phenotypic elements—root presence/absence (E1), canal root presence/absence (E2), canal location (E3), external root morphology (E4), and

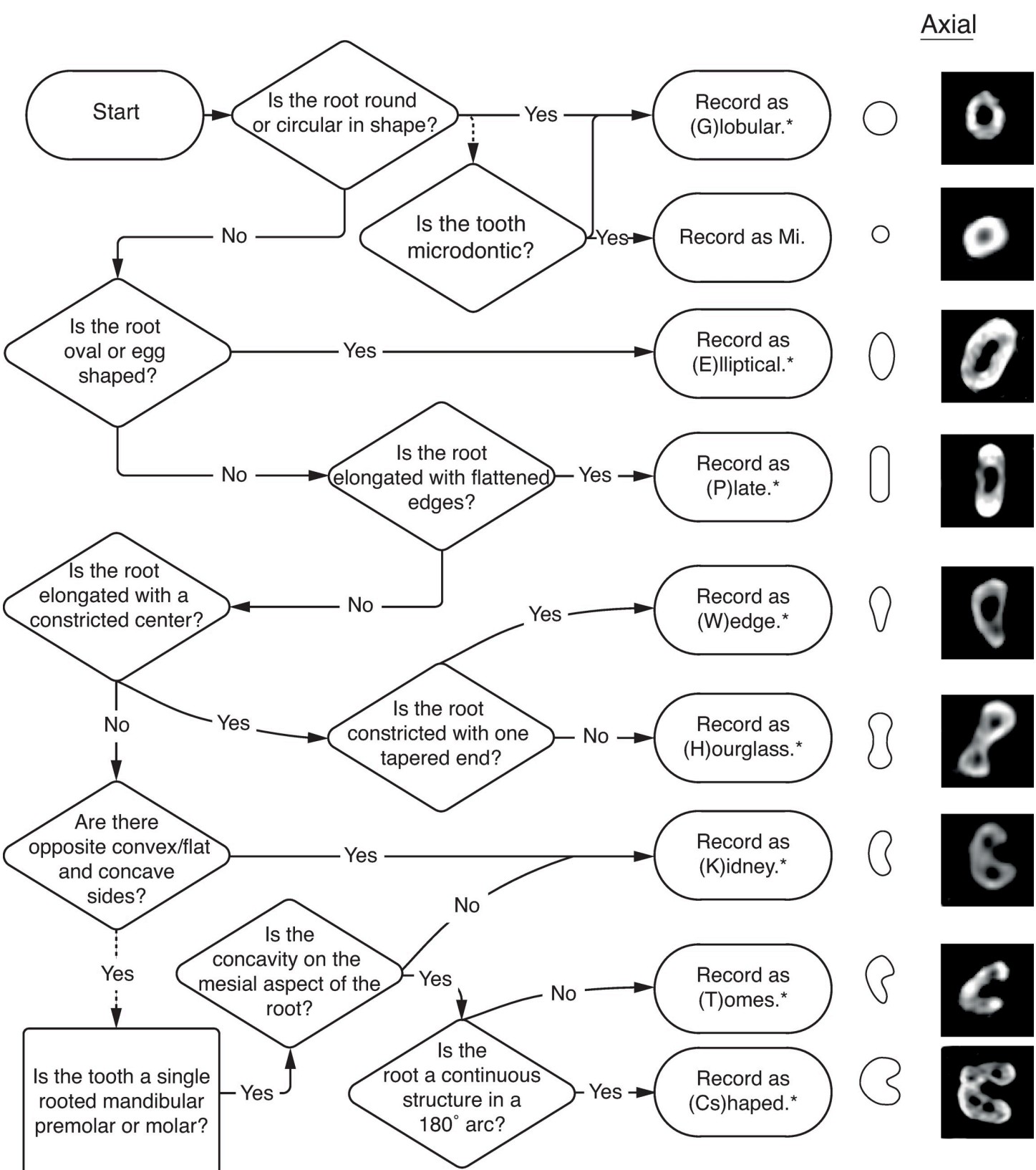

**Fig 16. Flow chart for determining and recording phenotype element 4—external root morphology.** *if root is bifurcated, append morphology with Bi. Ex: P = plate, PBi = plate-bifurcated. Right: axial CT slices showing external root morphologies.

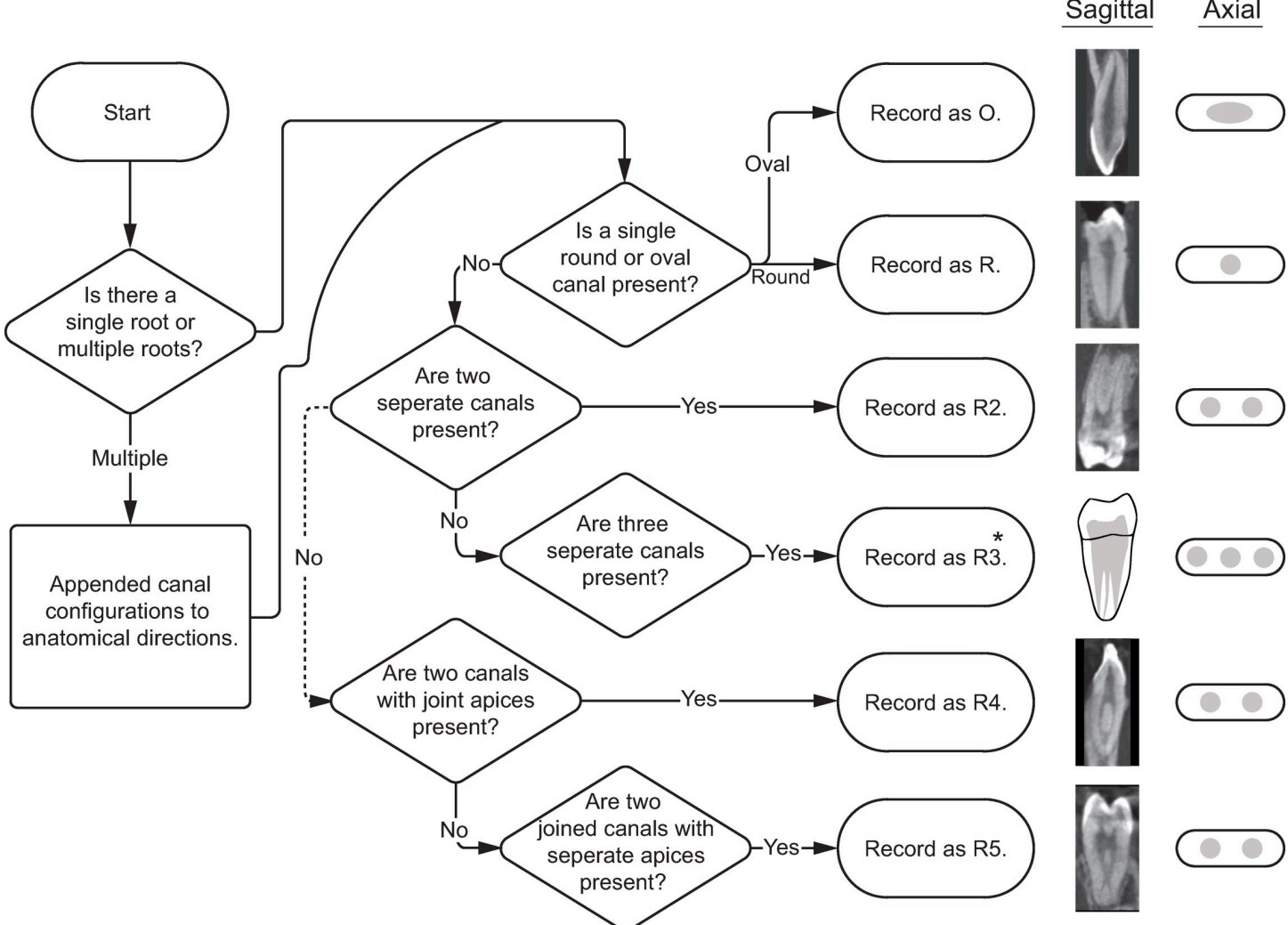

**Fig 17. Flow chart for determining and recording phenotype element 5—canal morphology and configuration.** Right: sagittal CT slices showing canal morphologies.
*Because the R3 variant does not appear in this sample, the sagittal slice is represented by an illustration.

canal morphology and configuration (E5), allow for independent categorization, so that phenotypes can be put together combinatorially, or treated as individual components–for example, using just external root morphology. Although constructed for recent human variation, the system can be extended across extant and fossil hominids, providing an additional tool for reconstructing evolutionary history, as well be used to map geographical patterns among contemporary human populations. Its broader applicability will be dependent upon an expansion in the number of scans available; while this is increasingly the case for fossil hominins, more regular scanning of more recent samples will be essential for studies of human diversity.

The advantages of this system, in addition to its universality, is that it allows for relatively simple qualitative and quantitative analysis. This is important, as there is increasing interest in mapping human diversity in different ways, using quantitative techniques [87–89]; the abundance of dental remains provides an additional source of information. In addition, there is growing interest among geneticists to map phenotypic variation against genetic variation [90], and to develop a better understanding of genotype-phenotype relationships. As teeth are

**Fig 18. Flow chart for determining and recording phenotype element 5—canal morphology and configuration (isthmus canals).** Illustrations show external root morphologies including C-shaped root variants. Canal shape/configuration is in gray.

generally to be considered strongly influenced by their genetic components [91, 92], they are an ideal system for testing these relationships. It may be the case that different phenotypes behave differently across populations, and so tooth roots can become part of phenotype-genotype comparisons. Such comparisons can be either phenetic, or phylogenetic, as the coding system is entirely suitable for cladistic analysis.

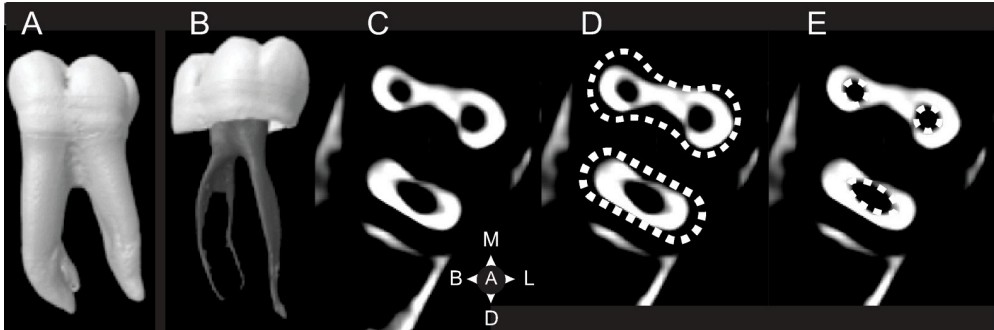

**Fig 19. Five phenotypic elements of a lower left 1st mandibular molar (RM$_1$-R2-C3-M2D1-MHDP-MR2DO).** A. E1—Root presence/absence; B. E2—Canal presence/absence, C. E3—Canal location, D. E4—Canal morphology, E. E5 —Canal shape. Images A and B from the Root Canal Anatomy Project https://rootcanalanatomy.blogspot.com/ (accessed 10 March 2021).

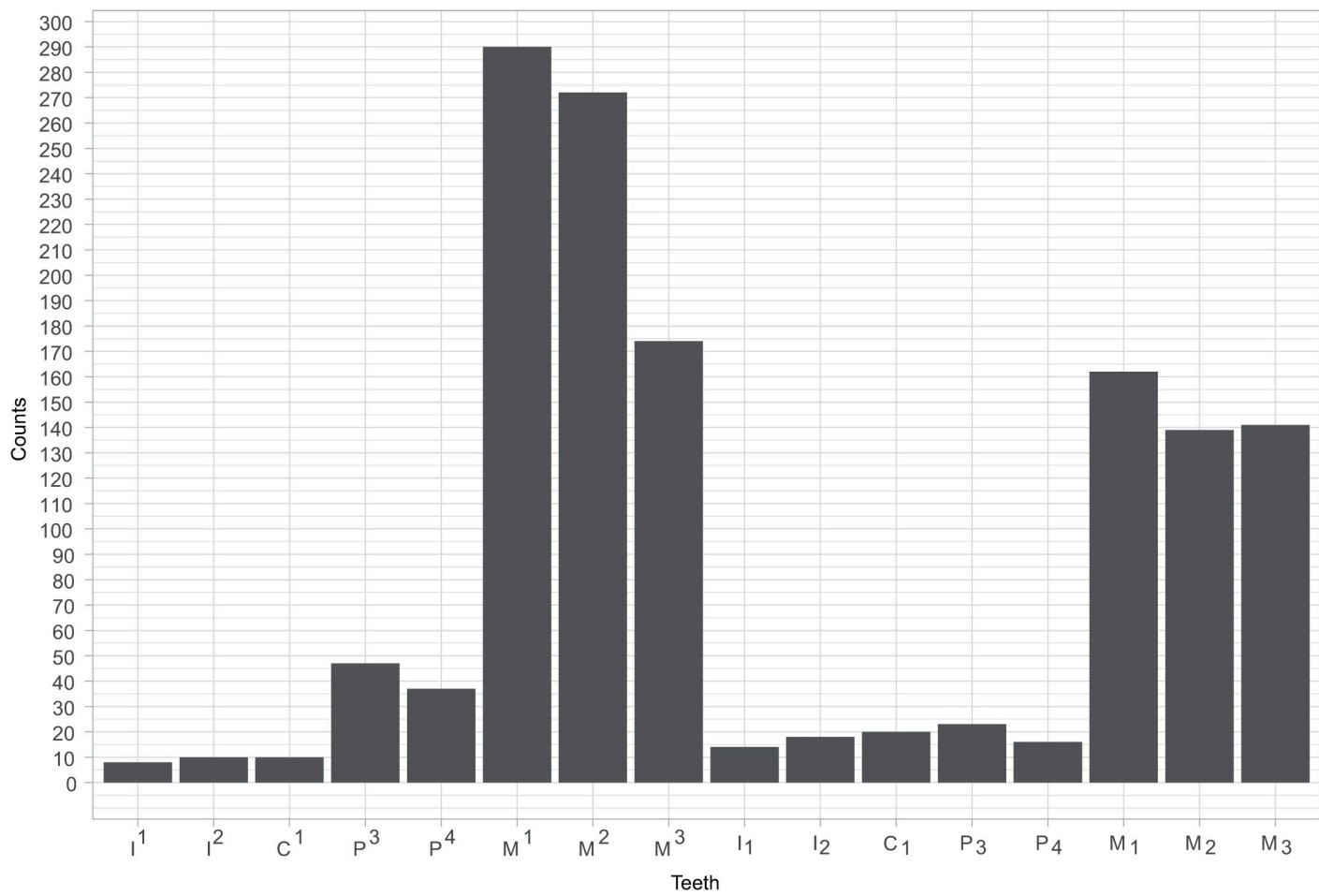

**Fig 20. Number of phenotypes in individual teeth.**

The second element relates to morphospace, an increasingly utilized concept in evolutionary biology [93, 94]. The morphospace is the total available forms that a phenotype can take, limited by physical or biological properties. Evolution is, in a sense, following paths in morphospace [95]. The phenotypic set is that part of the morphospace that is actually occupied. The system proposed here has explored the available morphospace for human tooth roots and has provided a series of elements that describe it. There are a very large number of possible phenotypes under this system. In principle, the total number is combinatorial product of the five phenotypic elements and their potential states, although in practice the number would be much smaller due to functional and physical constraints. In the relatively large sample utilized for this study there are about 874 observable individual tooth phenotypes–in other words a

**Table 11. Changing one element results in phenotype permutations in single-rooted teeth.**

| E1 | E2 | E3 | E4 | E5 | Code |
|----|----|----|----|----|------|
| R1 | C1 | A | **P** | O | R1-C1-A-P-O |
| R1 | C1 | A | **E** | O | R1-C1-A-E-O |
| R1 | C1 | A | **W** | O | R1-C1-A-W-O |
| R1 | C1 | A | **K** | O | R1-C1-A-K-O |

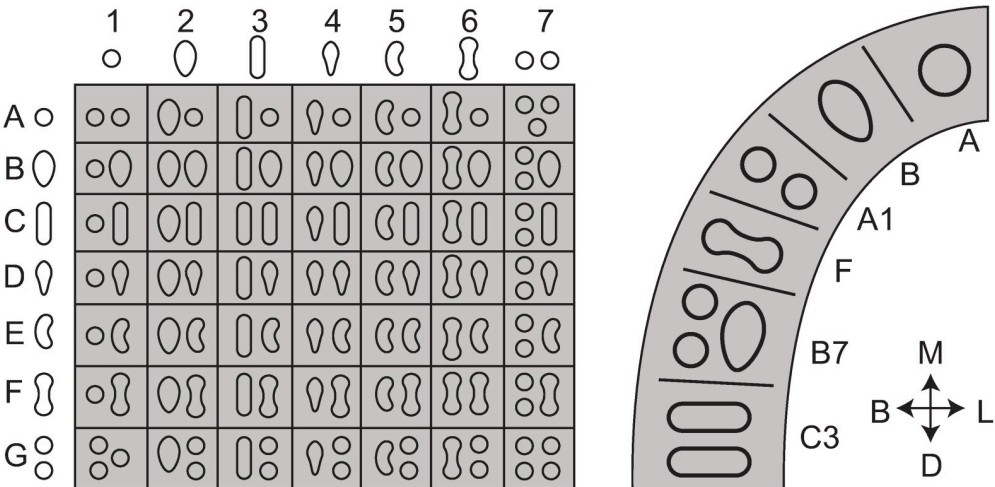

**Fig 21. Variation in the tooth root complex.** Left—Combinations of individual root types form multiple root complexes (e.g., C3 = one tooth with two plate shaped roots). Right—multiple root forms can appear in the tooth row. The left panel shows the range of possible combinations, while the right provides an example.

small proportion of possible ones. What is critical here is that the proposed method allows the realized and potential phenotypic sets of dental roots to be determined and analysed in potential evolutionary, developmental and functional contexts.

There is a bit of redundancy of information in this system. For example, R2-C3-M2D1-MHDP-MR2DO can be shortened to MHR2-DPO without loss of information. MHR2 describes a mesial (M) root (1 root) that is hourglass (H) shaped with two round (R2) canals (C), while DPO describes a distal (D) root (1 root) that is plate (P) shaped, with one ovoid (O) canal (C). However, there are several issues with this shorter version. The first is that this system was designed to record phenotype elements individually or in combination. MHR2 describes what is potentially a single rooted tooth or could be a four rooted tooth. R2 indicates that the root complex is two-rooted, as does M2D1, MHDP, and/or MR2DO. The second is human and computer readability. For a human, R2-C3-M2D1-MHDP-MR2DO is easier to read and understand than MHR2-DPO. For a computer, R2-C3-M2D1-MHDP-MR2DO allows easy separation and/or recombination of elements for analysis. The third is that not all users will have visual access to all elements within a root complex. It might be lack of equipment (radiography) or missing teeth. Thus, the system is also designed to capture the most information available to the user. Although there is a level of redundancy, the system is optimized for human and machine reading.

Future studies will benefit from the use of μCT rather than CT scans. While the resolution of CT scans is sufficient for this study as designed, they are not optimal for the segmentation of internal structures such as dentine and accessory canals. It is possible that both of these structures contain additional phenotypic information that can be quantified/qualified and appended to the system presented here. Additionally, segmentation of primary canal structures will increase the accuracy of morphological classification, allow for accurate metric analysis of internal and external structures, and help reduce any observation error due to misalignment of teeth against the anatomical plane.

Finally, for the method to be worthwhile, it is necessary for it to be useful in relation to current hypotheses and research foci. Four are immediately apparent. First, current interest in the role of dispersals, not just the initial one from Africa [96–98], but also the increasing genetic

evidence for multiple later regional dispersals means that finding ways of linking the palaeoanthropological and archaeological record to the inferred genotypes requires diverse phenotypes, and methods such as this will be required [99–102]. The second is in terms of earlier phases of human evolution; with the current evidence for interbreeding across hominin taxa [103], it is necessary to have appropriate phenotypic systems–and roots are likely to be a good one–to tease out the phenotypic effects in such admixture [104, 105]. Third, there is considerable interest in modularity and integration in evolution, and the modular approach adopted here may provide a suitable model system for exploring these issues [106, 107]. And finally, biomechanical and spatial studies of the hominid masticatory system can draw quantitative functional and dietary inferences from root and canal number and morphology [108–111].

## Conclusions

Compared to tooth crowns, tooth roots have received little attention in evolutionary studies. Novel technologies have increased the potential for exploiting variation in root morphology, and thus increased their importance as phenotypes. This paper presents a novel method for defining and analysing the morphospace of the human tooth-root complex. The five phenotypic elements of the system root presence/absence (E1), canal root presence/absence (E2), canal location (E3), external root morphology (E4), and canal morphology and configuration (E5), were designed to: 1) identify the elements that best describe variation in root and canal anatomy, 2) create a typology that is modular in nature and can be appended for undocumented morphotypes, and 3) is applicable to hominoids. The system will provide a basis for future research in human evolution, human genotype-phenotype investigations, and the functional biology of the human masticatory system.

## Supporting information

**S1 Table. List of individuals used in this study.**
(DOCX)

**S2 Table. Observer error test.**
(DOCX)

## Acknowledgments

We would like to thank Drs. Marta Miraźon-Lahr, Francis Rivera, and Lynn Copes for access to, and permission to use their CT scan collections. We would also like to thank James Clark for independently testing our system.

## Author Contributions

**Conceptualization:** Jason Gellis, Robert Foley.

**Data curation:** Jason Gellis.

**Formal analysis:** Jason Gellis.

**Investigation:** Jason Gellis.

**Methodology:** Jason Gellis, Robert Foley.

**Supervision:** Robert Foley.

**Validation:** Jason Gellis.

**Visualization:** Jason Gellis, Robert Foley.

**Writing – original draft:** Jason Gellis, Robert Foley.

**Writing – review & editing:** Jason Gellis, Robert Foley.

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
