## [Decision Letter · Decision Letter 0]

15 Jun 2021

PONE-D-21-14806

A novel system for classifying tooth root phenotypes

PLOS ONE

Dear Dr. Gellis,

Thank you for submitting your manuscript to PLOS ONE. After careful consideration, we feel that it has merit but does not fully meet PLOS ONE’s publication criteria as it currently stands. Therefore, we invite you to submit a revised version of the manuscript that addresses the points raised during the review process.

You should give special attention to the following points:

1. The comments of Reviewer 1.

2. The observer error is important for readers to assess the ease in using the system.

2. The slice thickness query of Reviewer 2.

3. Moving the anatomical description section as suggested by Reviewer 2.

Additional corrections:

p. 3 line 52 'a world frequency' does not have a range; rephrase

p. 19, line 415: the first 'their' should be 'therefore'.

p. 21, line 449 has grammatical errors and missing words; rephrase

p, 25 line 584 phenotype permutations, not 'phenotypes'

We look forward to receiving your revised manuscript.

Kind regards,

Lynne A Schepartz

Academic Editor

PLOS ONE

Journal Requirements:

2. Please declare author Jason Gellis's former affiliation with PLOS ONE as a competing interest.

4. We note you have included two tables which you refer in the text of your manuscript, however both are labelled as Table 10. Please ensure that you label each Table by a separate number in the title and also cite the relevant table number in your text; if accepted, production will need this reference to link the reader to each Table.

5. Please include captions for *all* your Supporting Information files at the end of your manuscript, and update any in-text citations to match accordingly. Please see our Supporting Information guidelines for more information: http://journals.plos.org/plosone/s/supporting-information.

Reviewers' comments:

Reviewer's Responses to Questions

**Comments to the Author**

1. Is the manuscript technically sound, and do the data support the conclusions?

Reviewer #1: Yes

Reviewer #2: Partly

2. Has the statistical analysis been performed appropriately and rigorously? 

Reviewer #1: Yes

Reviewer #2: No

3. Have the authors made all data underlying the findings in their manuscript fully available?

Reviewer #1: Yes

Reviewer #2: Yes

4. Is the manuscript presented in an intelligible fashion and written in standard English?

Reviewer #1: Yes

Reviewer #2: No

5. Review Comments to the Author

Reviewer #1: Thanks for inviting me to review this work on a new scoring system of root variation in modern humans. The authors thoroughly stated the needs of developing a new system to capture maximally the large degree of variation presented by the dental roots, and they did provided a useful system, covering five key elements of external and internal morphologies. However, I’m worried that the orientation of each tooth, although the jaw was placed in an anatomical position, might affect the score of external and internal morphologies. For example, a round canal cross-section might be observed in an elliptical shape if an oblique slice was obtained. A completely-quantitative method, in 3D context, might overcome all the problems of qualitative scoring system. I also have two very minor comments as listed below.

1) In the legend of Figure 3, which level did the author get the cross-section? Indicate what "E, G, H, and P" means in the figure legend.

2) Figure 4 and 9 are the same, except for the labels.

Reviewer #2: The idea presented is novel and very interesting. There are a few areas which require attention:

In general there are a significant number of figures, which ideally could be condensed to reflect the most pertinent figures in the study. As this is a scientific article all references to "we" should be removed,

The introduction would benefit from including a short section discussing the potential genetic abnormalities associated with root morphology, particularly considering that the "normal" morphology of the roots is the subject under investigation.

Background: Pg 3, line 47, reference numbers can be condensed to reflect 12-26. In the legend for figure 1, line 77, specify which molars are being referenced to keep consistency with the rest of the legend. Pg. 5, Line 124, replace "Though" with "While", Line 127, insert references, particularly if you are talking about descriptions which appear in the literature. While considering the external root morphology was the possibility of enamel pearls considered?

Materials and methods: Please change specimens to individuals throughout. The SI and AMNH samples were scanned with a slice thickness of 1.0mm and reconstrcted while the DC sample had a slice thickness of 0.6mm and no reconstruction. Do you think the difference in slice thickness may have resulted in a loss of observed detail in the scans? No exclusion criteria is mentioned and should be included.

The tenses in this section need revision. As the article happens after the study the article should be written in the past tense. Please correct throughout. Pg 11. line 312, Anatomical descriptions should come earlier, preferrably straight after the CT section. No mention of observer error is mentioned at any point, this is problematic considering the proposal of novel system for classifying teeth.

Results: Pg 12 Lines 331 to 348 should ideally come under the materials and methods as it is an admixture of a sample description as well as data analysis. The results section requires extensive revision given that across various points in the results, i.e. Table 6 line 404, there appears to be a mixture of result reporting and discussion of the results. Strictly speaking the results section is where we present the observations made in the current study and the discussion section is where the current study's results are put into the context of the broader literature and any variances are discussed further. The redundancy of information section, line 547 should not be in the results section and ideally should form part of the discussion section

Discussion: The inclusion of tables and figures in the discussion section is irregular.

6. PLOS authors have the option to publish the peer review history of their article (what does this mean?). If published, this will include your full peer review and any attached files.

Reviewer #1: No

Reviewer #2: No

---

## [Author Response · Author response to Decision Letter 0]

10 Sep 2021

We have uploaded a document - Response to Reviewers, which covers all issues for mauscript revision.

---

## [Editor Report · Decision Letter 1]

17 Sep 2021

A novel system for classifying tooth root phenotypes

PONE-D-21-14806R1

Dear Dr. Gellis,

We’re pleased to inform you that your manuscript has been judged scientifically suitable for publication and will be formally accepted for publication once it meets all outstanding technical requirements.

Kind regards,

Lynne A Schepartz

Academic Editor

PLOS ONE
---

## [Editor Report · Acceptance letter]

18 Oct 2021

PONE-D-21-14806R1 

A novel system for classifying tooth root phenotypes 

Dear Dr. Gellis:

I'm pleased to inform you that your manuscript has been deemed suitable for publication in PLOS ONE. Congratulations! Your manuscript is now with our production department. 

Kind regards, 

on behalf of

Dr. Lynne A Schepartz 

Academic Editor

PLOS ONE